# Unisoma: A Unified Transformer-based Solver for Multi-Solid Systems

**Shilong Tao** [1]  **Zhe Feng** [1]  **Haonan Sun** [1]  **Zhanxing Zhu** [2]  **Yunhuai Liu** [1]

## Abstract

Multi-solid systems are foundational to a wide range of real-world applications, yet modeling their complex interactions remains challenging. Existing deep learning methods predominantly rely on implicit modeling, where the factors influencing solid deformation are not explicitly represented but are instead indirectly learned. However, as the number of solids increases, these methods struggle to accurately capture intricate physical interactions. In this paper, we introduce a novel explicit modeling paradigm that incorporates factors influencing solid deformation through structured modules. Specifically, we present Unisoma, a unified and flexible Transformer-based model capable of handling variable numbers of solids. Unisoma directly captures physical interactions using contact modules and adaptive interaction allocation mechanism, and learns the deformation through a triplet relationship. Compared to implicit modeling techniques, explicit modeling is more well-suited for multi-solid systems with diverse coupling patterns, as it enables detailed treatment of each solid while preventing information blending and confusion. Experimentally, Unisoma achieves consistent state-of-the-art performance across seven well-established datasets and two complex multi-solid tasks. Code is avaiable at https://github.com/therontau0054/Unisoma.

## 1. Introduction

Multi-solid (rigid and deformable) systems play an essential role in a broad range of real-world scenarios, such as industrial manufacturing (Khan & Turowski, 2016), mechanical manipulation (Mason, 2001), and aerospace (Mouritz, 2012). For example, multi-point metal stamping (Li et al., 2002)

[1]School of Computer Science, Peking University, Beijing, China [2]School of Electrical and Computer Science, University of Southampton, UK. Correspondence to: Yunhuai Liu <yunhuai.liu@pku.edu.cn>, Zhanxing Zhu <z.zhu@soton.ac.uk>.

*Proceedings of the 42$^{nd}$ International Conference on Machine Learning*, Vancouver, Canada. PMLR 267, 2025. Copyright 2025 by the author(s).

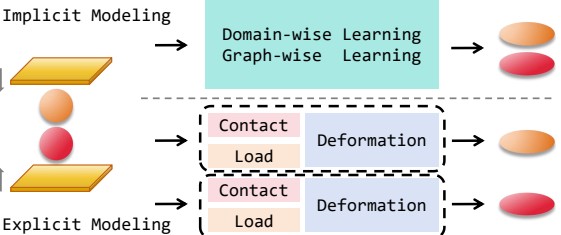

*Figure 1.* Implicit Modeling and Explicit Modeling. Yellow planes are rigid solids. Orange and red balls are deformable solids.

in industry, robotic gripping (Zhang et al., 2020), and composite materials research (Clyne & Hull, 2019) are typical application scenarios. From a scientific standpoint, multi-solid systems can often be formulated as high-dimensional, nonlinear Partial Differential Equations (PDEs). However, the large solution space, strict contact conditions, and complex dynamics pose significant computational challenges for traditional numerical methods (e.g., Finite Element Methods (Sifakis & Barbic, 2012) and Material Point Methods (Sulsky et al., 1995)), making them difficult to meet real-time or high-fidelity requirements (Umetani & Bickel, 2018). Recently, Deep Learning techniques have been increasingly adopted in scientific computing, showing promise for various physical simulations (Cai et al., 2021; Lu et al., 2021; Li et al., 2020a). Despite these advances, most research to date has focused on few-solid systems or other physical scenarios such as fluids. There remains a notable gap in applying deep learning to multi-solid systems, and a systematic approach capable of characterizing *complex physical interactions among multiple solids* is still underexplored.

Multi-solid systems encompass multiple rigid and deformable solids with distinct mechanical properties (e.g., elastic, plastic, hyperelastic). The motion of these solids and their contact interactions induce forces (also referred to as loads), thereby driving deformations and changes in the physical quantities of deformable solids, which are our primary focus. In these systems, diverse material behaviors, contact phenomena, and exerted loads converge into complex physical interactions, which in turn lead to intricate deformation behaviors (Braess, 2001). Consequently, *how to effectively distinguish and capture these interactions* is the key challenge in multi-solid analysis.

Presently, research specifically dedicated to multi-solid sys-

tems remains relatively scarce. In addressing solid problems, data-driven approaches predominantly rely on **implicit modeling**, where the factors that influence the deformation of deformable solids are not explicitly structured but instead implicitly captured through the overall learning process. These methods can be broadly categorized into two types. As shown in the upper branch of Figure 1, the first category (Raissi et al., 2019; Li et al., 2020a) is domainwise, treating the entire computational domain as input and merging all solids into one large PDE problem. Then it is solved through Physics-Informed Neural Networks (PINNs) (Cai et al., 2021) or Neural Operators (Boullé & Townsend, 2023). Although this approach simplifies setup and ensures global consistency, it can become computationally expensive and lacks fine-grained insight into individual solids. As the number and diversity of solids increase, the effective capture of intricate physical interactions becomes increasingly challenging, restricting its scalability. The second category (Sanchez-Gonzalez et al., 2018; 2020a) is graph-wise, using graph topologies to represent each solid as a subdomain and dynamically add or remove edges between solids based on a physical distance threshold. The GNN-based message passing (Gilmer et al., 2017) is then utilized to transmit and aggregate information between nodes. In this process, the edges merely act as conduits for information flow, and the interactions influencing the deformation of solids are still implicitly learned through the node-level propagation. However, only relying on independent edges to learn physical interactions between different solids may introduce blending and ambiguity (Yifan et al., 2020). In addition, the local nature of graph kernels limits its efficacy for long-range predictions involving long time span (Li et al., 2024a).

As shown in the lower branch of Figure 1, unlike previous approaches, we propose to solve multi-solid systems within an **explicit modeling** paradigm, which directly models the factors (such as contact constraints and loads) that influence the deformation of solids through the model structure. This design is especially well-suited to multi-solid scenarios where coupling patterns are diverse, allowing for tailored treatment of different pairs with more fine-grained modeling. In this way, we can distinguish and capture essential physical interactions more controllably and accurately. It avoids the issues of information confusion and the difficulty in fully learning complex physical interactions in multi-solid scenarios that are common in implicit modeling methods.

Technically, we propose **Unisoma**, a unified and flexible Transformer-based (Vaswani, 2017) framework with explicit modeling capable of handling variable numbers of solids and two important tasks (long-time prediction and autoregressive simulation) in the multi-solid scenarios, as the first attempt of the explicit modeling to our best knowledge. Specifically, we define a deformation triplet consisting of a deformable solid, an equivalent load, and an equivalent

contact constraint to describe the deformation of a single deformable solid in the system. We first utilize distinct contact modules to capture each pair of contact constraints in the system. Then we propose an adaptive interaction allocation mechanism to compute the individual equivalent load and contact constraint for each deformable solid. Afterwards, for each deformable solid, we adopt a deformable module to model the influence of equivalent load and contact constraint on its deformation under deformation triplet relationship. Unisoma achieves state-of-the-art performance on extensive evaluations. Our main contributions are as follows:

- We propose to solve multi-solid systems in an explicit modeling paradigm. This kind of paradigm brings a high degree of flexibility, enabling effective control and capture of essential physical interactions.

- We design Unisoma, a unified Transformer-based model with an explicit modeling approach capable of flexibly handling varying numbers of solids. Unisoma captures and integrates physical interactions with explicitly structured modules, and learn the deformation under the deformation triplet relationship.

- We evaluate Unisoma on two multi-solid-related tasks across seven datasets spanning different complexity, comparing it against over ten advanced deep learning models. Unisoma consistently achieves superior performance improvements under all test scenarios.

## 2. Related Work

Currently, there is a relative scarcity of research focused on multi-solid systems. For solid problems, data-driven frameworks are generally divided into two paradigms.

**Implicit Modeling** In this paradigm, the factors that influence the deformation of solids are not explicitly structured but instead implicitly learned from data. Methods belonging to this paradigm can be classified into two categories. The first domain-wise category treats the entire computational domain as input, merging all objects into one PDE problem. In the solid field, PINNs have been widely applied (Haghighat et al., 2021; Rodriguez-Torrado et al., 2021; Okazaki et al., 2022), often with elaborately designed loss functions (Bai et al., 2023; Chiu et al., 2022) and domainwise techniques (Diao et al., 2023; Li et al., 2023a). Neural Operators, such as FNO (Li et al., 2020a) and its variants (Gupta et al., 2021; Deng et al., 2024), have demonstrated success in simple solid problems with regular geometries. Many variants (Li et al., 2020b; 2024a) are proposed to tackle irregular geometries. In particular, the geometric Fourier transform proposed in Geo-FNO (Li et al., 2023b) is vastly used, which projects the irregular geometry into uniform latent structure. Transformer-based models (Vaswani,

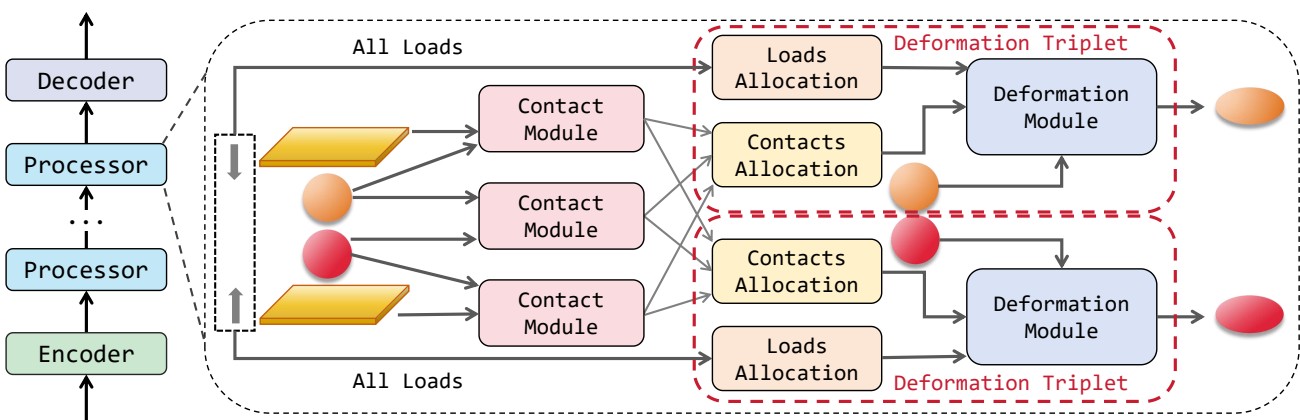

*Figure 2.* The overview of Unisoma. The contact module and deformation module are attention-based. The loads allocation module and contacts allocation module deploy adaptive interaction allocation mechanism.

2017) have gained traction in continuum mechanics, utilizing advanced attention mechanisms (Liu et al., 2024; Li et al., 2024b; Wu et al., 2024; Xiao et al., 2024; Li et al., 2024c) and physics-informed inductive biases (Li et al., 2023d; Hao et al., 2023; Shao et al., 2022). However, all these models are primarily evaluated on simple few-solid scenarios. When tackling multi-solid systems, they struggle to effectively capture distinct physical interactions due to the lack of fine-grained insight into individual solids, particularly as the number and diversity of solids increase.

The second graph-wise category encodes the physical objects as graphs, and leverage GNNs (Scarselli et al., 2008) to capture the interactions in a local neighbor with radius threshold. In the solid field, GNN-based simulators have been vastly used in particle-based simulations (Li et al., 2019; Sanchez-Gonzalez et al., 2020b) and mesh-based simulations (Weng et al., 2021; Han et al., 2022; Cao et al., 2023). MGN (Pfaff et al., 2021) utilizes message passing blocks (Gilmer et al., 2017) to learn the dynamics of physical systems. Many subsequent works have enhanced performance through hierarchical structures (Grigorev et al., 2023; Yu et al., 2024; Fortunato et al., 2022), physics-informed guidance (Würth et al., 2024; Perera & Agrawal, 2024) and diverse geometrical information (Linkerhägner et al., 2023; Allen et al., 2022). However, when independent edges are used as the learning medium, it may cause ambiguity and blending of the physical interactions between the solids. Additionally, graph-based simulators still fall short in learning global interactions and struggle to address long-time predictions involving long time span (Li et al., 2024a).

**Explicit Modeling** In contrast, we propose an explicit modeling paradigm, explicitly incorporating factors influencing solid deformation through specialized modules. Technically, we propose Unisoma, a unified and flexible Transformer-based framework, as the first attempt of the explicit model-

ing for multi-solid systems to our best knowledge. It explicitly models physical interactions with structured modules and learns deformation through deformation triplet relationship. Compared to implicit modeling, explicit modeling is better suited for multi-solid systems with diverse coupling patterns, as it allows fine-grained treatment of each solid and avoids information blending and confusion.

## 3. Method

**Problem Setup** Consider a system with multiple objects composed of $N^d$ deformable solids $\mathbf{u}^d = \{\mathbf{u}_i^d \in \mathbb{R}^{N_i^d \times C_d}\}_{i=1}^{N^d}$ with different material properties, $N^r$ rigid solids $\mathbf{u}^r = \{\mathbf{u}_i^r \in \mathbb{R}^{N_i^r \times C_r}\}_{i=1}^{N^r}$, and $N^f$ acting forces (loads) $\mathbf{u}^f = \{\mathbf{u}_i^f \in \mathbb{R}^{N_i^f \times C_f}\}_{i=1}^{N^f}$ yielded by the movement of solids in the system. They are all mesh points described in the Lagrangian view. A point of a deformable solid $\mathbf{d}_i$ or rigid solid $\mathbf{r}_i$ is recorded as a vector of length $C_d$ or $C_r$, composed of 3D space coordinates and its properties (like Young's Modulus, Poisson's Rate, Stress and Friction). A point of load $\mathbf{f}_i$ is a vector of length $C_f$ recording the 3D space coordinates of the solid from which it originated, and its next movement or position. For example, let $\mathbf{u}_{i,t}$ and $\mathbf{u}_{i,t+1}$ denote the current and next state of a moving solid, the load is represented as $\text{Concat}(\text{Coord}(\mathbf{u}_{i,t}), \text{Coord}(\mathbf{u}_{i,t+1}) - \text{Coord}(\mathbf{u}_{i,t}))$ or $\text{Concat}(\text{Coord}(\mathbf{u}_{i,t}), \text{Coord}(\mathbf{u}_{i,t+1}))$, where $\text{Coord}(\cdot)$ indicates the 3D space coordinates of the solid.

We consider two types of tasks: 1) **Long-Time Prediction** (Raissi et al., 2019; Li et al., 2020a; Wu et al., 2024): Directly estimate target physical quantities (like geometries and inner stress) of deformable solids with long time duration and without intermediate process. It only inferences one time as: $P(\hat{\mathbf{u}}_{i,t+T}|\mathbf{u}_{i,t})$, where $T$ spans many time steps. 2) **Autoregressive Simulation** (Pfaff et al., 2021; Yu et al., 2024; Li et al., 2019): Autoregressively

simulate the deforming process of deformable solids with small time increments. It inferences many times as: $P(\hat{\mathbf{u}}_{i,t+1}|\mathbf{u}_{i,t}), P(\hat{\mathbf{u}}_{i,t+2}|\hat{\mathbf{u}}_{i,t+1}), \cdots, P(\hat{\mathbf{u}}_{i,t+T}|\hat{\mathbf{u}}_{i,t+T-1})$.

### 3.1. Model Overview

The overview of Unisoma is illustrated in Figure 2, following the "encoder-processor-decoder" fashion (Battaglia et al., 2018). First, instead of processing the multiple solids in the original input domain, we embed each object into edge augmented physics-aware tokens (Section 3.2). We then adopt stacked processors to explicitly model physical interactions among solids, including equivalent loads, equivalent contact constraints, and solid deformation (Section 3.3). Throughout the process, tokens representing rigid solids remain unchanged, while those representing deformable solids are updated. This reflects the physical principle that rigid solids are non-deformable. Finally, the outputs of the final processor are mapped back to the original domain, resulting in the predicted outcomes. Now we elaborate on each key component of Unisoma.

### 3.2. Encoder: Edge Augmented Physics-Aware Tokens

Originally, the objects in the physical domain are described as a large number of mesh points. We first need to embed them into tokens. A naive approach treats each point as a token, unfortunately leading to high computational cost and redundant representation (Yu et al., 2022). Transolver (Wu et al., 2024) proposes the concept of slice, which assigns mesh points with similar physical states to the same learnable slice token, enabling attention to capture intrinsic physical correlations more effectively. However, they only consider point-level features and spatially aggregate points in the whole domain. This leads to the loss of local relationships of mesh points, which are important to model local interactions like contact and local deformation. Therefore, we incorporate the mesh edges into embedding and propose *Edge Augmented Physics-Aware Tokens*.

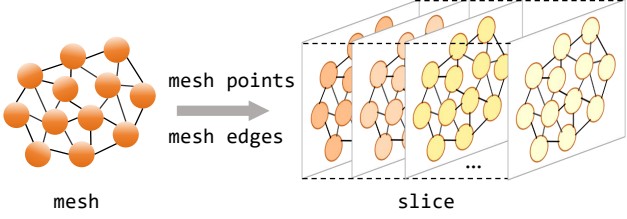

*Figure 3.* Embedding mesh points and mesh edges into slices.

For each object, we first utilize the $k$-nearest neighbor (kNN) method to construct the edge set $E$ based on input mesh points. The edge attribution is the relative distance of the connected neighbors. Let $\mathbf{x} \in \mathbb{R}^{N \times C}$ be the deep features of an object with $N$ mesh points projected by linear layers

(i.e., $\mathbf{x} = \text{Linear}(\mathbf{u})$) and $\mathbf{e}_{p,q} \in \mathbb{R}^{1 \times C}$ be the deep feature of edge connecting mesh points $\mathbf{x}_p$ and $\mathbf{x}_q$. The edge augmented physics-aware tokens are formalized as:

$$
\begin{aligned}
\mathbf{w}_i &= \{\text{Softmax}(\text{Linear}(\mathbf{x}_i))\} \\
\mathbf{w}_{(p,q)}^e &= \{\text{Softmax}(\text{Linear}(\mathbf{w}_p + \mathbf{w}_q))\} \\
\mathbf{z}_j &= \frac{\sum_{i=1}^{N} \mathbf{s}_{j,i} + \gamma \sum_{\mathbf{e}_{p,q} \in E} \mathbf{s}_{j,(p,q)}^e}{\sum_{i=1}^{N} \mathbf{w}_{i,j} + \gamma \sum_{\mathbf{e}_{p,q} \in E} \mathbf{w}_{(p,q),j}^e} \\
&= \frac{\sum_{i=1}^{N} \mathbf{w}_{i,j}\mathbf{x}_i + \gamma \sum_{\mathbf{e}_{p,q} \in E} \mathbf{w}_{(p,q),j}^e \mathbf{e}_{p,q}}{\sum_{i=1}^{N} \mathbf{w}_{i,j} + \gamma \sum_{\mathbf{e}_{p,q} \in E} \mathbf{w}_{(p,q),j}^e}
\end{aligned} \quad (1)
$$

Here, $\mathbf{s} = \{\mathbf{s}_j \in \mathbb{R}^{N \times C}\}_{j=1}^{M}$ are termed slices with number of $M$, whose weights $\mathbf{w} \in \mathbb{R}^{N \times M}$ are yielded from deep features $\mathbf{x}$ by $\text{Linear}(\cdot)$ projection and $\text{Softmax}(\cdot)$ operation. The weight $\mathbf{w}_{i,j}$ indicates the degree that the $i$-th mesh point belongs to the $j$-th slice with $\sum_{j=1}^{M} \mathbf{w}_{i,j} = 1$. The mesh points with close geometry and physical features are more likely to be assigned to the same slice. As shown in Figure 3, we incorporate the mesh edges into the slice components $\mathbf{s}^e = \{\mathbf{s}_j^e \in \mathbb{R}^{|E| \times C}\}_{j=1}^{M}$, whose weights $\mathbf{w}^e \in \mathbb{R}^{|E| \times M}$ are derived from the weights of the connected points. The physics-aware tokens $\mathbf{z} = \{\mathbf{z}_j \in \mathbb{R}^{1 \times C}\}_{j=1}^{M}$ then are encoded by spatially weighted aggregation and normalization. Notably, although we utilize graph-like representation of the objects, it differs from GNN-based methods that capture interactions in the node level with edges. We transfer the graph into the slice domain and explicitly model physical interactions at the object level with attention mechanism.

*Remark* 3.1. *Why incorporating edge features by spatially weighted aggregation?* Firstly, the aggregation follows the transformation of the original slice operated on mesh points (Wu et al., 2024) and the Eq.(1) is a unified projection form including mesh points and edges. The Eq.(1) degenerates to the original slice if edge feature is not taken into account. Secondly, the weights of edge features are derived from the weights of mesh point features. This aggregation emphasizes point pairs that are connected by edges, thereby further enhancing the local information during the aggregation process. Normally, the number of edges is more than the number of points. To avoid the point features being inundated by edge features, we utilize the ratio of points and edges $\gamma = N/|E|$ for quality control.

The embedding is performed for each object instead of the holistic domain for flexibly explicit modeling. We denote the embedding tokens for deformable solids $\mathbf{u}^d$, rigid solids $\mathbf{u}^r$, and loads $\mathbf{u}^f$ as $\mathbf{d} = \{\mathbf{d}_i \in \mathbb{R}^{M \times C}\}_{i=1}^{N^d}$, $\mathbf{r} = \{\mathbf{r}_i \in \mathbb{R}^{M \times C}\}_{i=1}^{N^r}$, and $\mathbf{f} = \{\mathbf{f}_i \in \mathbb{R}^{M \times C}\}_{i=1}^{N^f}$, respectively.

### 3.3. Processor

After embedding these objects into edge augmented physics-aware tokens, we propose an extensible processor to learn

the physical interactions in an explicit manner. As shown in the right part of Figure 2, the design of the processor revolves around the deformation triplet. We separately consider the loads and contact constraints and then capture their effects on the deformable solids. First, we design a contact module to capture each pair of contacts that may occur in the system. Next, we propose an adaptive interaction allocation mechanism to learn the equivalent load and contact constraint for each individual deformable solid. Finally, we introduce a deformable module to characterize the influence of the equivalent load and the contact constraint on the deformation of the corresponding solid.

**Deformation Triplet** In a multi-solid system, the main factors influencing the deformation of a deformable solid are loads and contact constraints (Shabana, 2020; Wriggers & Laursen, 2006; Bathe, 2006). Loads are the forces that act on the system, causing stress and strain on the deformable solids, thereby inducing deformation. Contact constraints refer to the geometric conditions that a solid experiences when in contact with other solids. These constraints alter the force conditions experienced by the solid, thereby affecting its deformation. Therefore, the key insight of explicitly modeling the two factors in deformation guides us to propose a triplet relationship $(\mathbf{d}_i, \overline{\mathbf{f}}, \overline{\mathbf{c}})$ to represent the deformation of a single deformable solid $\mathbf{d}_i$. Here, $\overline{\mathbf{f}} \in \mathbb{R}^{M \times C}$ is the equivalent load, which represents the comprehensive effect of all loads in the system on the solid $\mathbf{d}_i$, and $\overline{\mathbf{c}} \in \mathbb{R}^{M \times C}$ is the equivalent contact constraint, comprehensively modeling all contacts experienced by the solid $\mathbf{d}_i$ in the system.

**Contact Module** A pair of contacts can occur between two solids. In a multi-solid system, multiple pairs of contacts may exist. It is hard to distinguish between these contact constraints if we merge all solids as one input (Haghighat et al., 2021; Li et al., 2020a). Therefore, we model each pair of possible contacts individually. As shown in the right part of Figure 2, for any two solids $\mathbf{g}_i$ and $\mathbf{g}_j$ that are likely to contact, where $\mathbf{g}_i, \mathbf{g}_j \in \mathbf{d} \cup \mathbf{r}$ and $1 \le i, j \le N^d + N^r$, we propose a contact module to capture the contact constraint $\mathbf{c}_k$ between them, formulated as follows:

$$\mathbf{Q}, \mathbf{K}, \mathbf{V} = \text{Linear}(\mathbf{g}_i + \mathbf{g}_j)$$
$$\mathbf{c}_k = \text{Softmax}\left(\frac{\mathbf{Q}\mathbf{K}^T}{\sqrt{C}}\right)\mathbf{V} \tag{2}$$

We add these two tokens as the input and employ the attention mechanism to capture the intricate contact constraint. Totally $N^c$ contact modules are arranged to learn $N^c$ pairs of contact constraints $\mathbf{c} = \{\mathbf{c}_k \in \mathbb{R}^{M \times C}\}_{k=1}^{N^c}$. For a system with different numbers of contact pairs, we can easily extend contact modules in the width direction. It brings vast flexibility and applicability for various multi-solid systems.

*Remark* 3.2. *Why is addition operation effective in slice domain?* Directly adding two high-dimensional features

representing different objects in the original input space is generally not effective. Unlike the residual connections (He et al., 2016), which are mainly designed to combine features of the same object, summing features from distinct objects can lead to significant information loss. Specifically, this operation causes an indiscriminate mixture of two feature spaces, obscuring the unique physical attributes of each object. In particular, tokens at corresponding positions may encode crucial physical properties, but their summation with unrelated tokens may distort these details. However, we argue that the addition operation is effective in slice domain. As mentioned before, we separately embed each object in the system into their own slice domains with size $M$. In fact, this separate embedding is a special case of the holistic embedding. Technically, for deep features $\mathbf{x}^\alpha$ and $\mathbf{x}^\beta$ of any two objects, we embed them into slice $\mathbf{z}^{\alpha\beta} \in \mathbb{R}^{M \times C}$ with slice weight $\mathbf{w}^{\alpha\beta} = \{\mathbf{w}_{1,j}^\alpha, \mathbf{w}_{2,j}^\alpha, \cdots, \mathbf{w}_{N^\alpha,j}^\alpha, \mathbf{w}_{1,j}^\beta, \mathbf{w}_{2,j}^\beta, \cdots, \mathbf{w}_{N^\beta,j}^\beta\}$:

$$\begin{aligned}\mathbf{z}_j^{\alpha\beta} &= \frac{\sum_{i=1}^{N^\alpha+N^\beta} \mathbf{w}_{i,j}^{\alpha\beta}\text{Concat}(\mathbf{x}^\alpha, \mathbf{x}^\beta)_i}{\sum_{i=1}^{N^\alpha+N^\beta} \mathbf{w}_{i,j}^{\alpha\beta}} \\ &= \frac{\sum_{i=1}^{N^\alpha} \mathbf{w}_{i,j}^\alpha \mathbf{x}_i^\alpha + \sum_{i=1}^{N^\beta} \mathbf{w}_{i,j}^\beta \mathbf{x}_i^\beta}{\sum_{i=1}^{N^\alpha} \mathbf{w}_{i,j}^\alpha + \sum_{i=1}^{N^\beta} \mathbf{w}_{i,j}^\beta}\end{aligned} \tag{3}$$

Following this formulation, embedding each object individually can be formulated as:

$$\begin{aligned}\mathbf{z}_j^\alpha &= \frac{\sum_{i=1}^{N^\alpha} \mathbf{w}_{i,j}^\alpha \mathbf{x}_i^\alpha + \sum_{i=1}^{N^\beta} \mathbf{w}_{i,j}^\beta \mathbf{x}_i^\beta}{\sum_{i=1}^{N^\alpha} \mathbf{w}_{i,j}^\alpha + \sum_{i=1}^{N^\beta} \mathbf{w}_{i,j}^\beta} \quad (\mathbf{w}^\beta = \mathbf{0}) \\ &= \frac{\sum_{i=1}^{N^\alpha} \mathbf{w}_{i,j}^\alpha \mathbf{x}_i^\alpha}{\sum_{i=1}^{N^\alpha} \mathbf{w}_{i,j}^\alpha}\end{aligned} \tag{4}$$

For clarity, we omit the mesh edges here, and the complete form can be found in Appendix B, where it still holds. Through Eq.(4), we can observe that the separate embedding is a special case that the slice weights for other objects are $\mathbf{0}$. This indicates that we build a pure slice domain on the holistic input domain but only projected by a single object. Furthermore, the slice $\mathbf{z}_j^{\alpha\beta}$ can be reformulated as follows:

$$\begin{aligned}\mathbf{z}_j^{\alpha\beta} &= \frac{\sum_{i=1}^{N^\alpha} \mathbf{w}_{i,j}^\alpha \mathbf{x}_i^\alpha + \sum_{i=1}^{N^\beta} \mathbf{w}_{i,j}^\beta \mathbf{x}_i^\beta}{\sum_{i=1}^{N^\alpha} \mathbf{w}_{i,j}^\alpha + \sum_{i=1}^{N^\beta} \mathbf{w}_{i,j}^\beta} \\ &= \frac{(\sum_{i=1}^{N^\alpha} \mathbf{w}_{i,j}^\alpha)\mathbf{z}_j^\alpha + (\sum_{i=1}^{N^\beta} \mathbf{w}_{i,j}^\beta)\mathbf{z}_j^\beta}{\sum_{i=1}^{N^\alpha} \mathbf{w}_{i,j}^\alpha + \sum_{i=1}^{N^\beta} \mathbf{w}_{i,j}^\beta} \\ &\approx \theta \mathbf{z}_j^\alpha + (1-\theta)\mathbf{z}_j^\beta\end{aligned} \tag{5}$$

Here, $\mathbf{z}^{\alpha\beta}$ can be seen as the composition of $\mathbf{z}^\alpha$ and $\mathbf{z}^\beta$ through coefficient $\theta$, referred to as *slice composition*. Accordingly, the operation in Eq.(4) is termed *slice decomposition*. We first construct multiple pure slice domains during

embedding. Through slice composition, we merge two slice domains that are contact-related. In practice, we adopt direct element-wise addition in Eq.(2) as a simple, parameter-free realization of this linear combination. This design achieves comparable performance to the learnable form, while reducing complexity. Although this simplified form does not perform explicit averaging (e.g.,$0.5(\mathbf{z}_j^\alpha + \mathbf{z}_j^\beta)$), the resulting features are subsequently processed by normalization and attention layers (e.g., in the contact module), which mitigates effects from scale differences. We then apply attention mechanism to capture the physical interaction within the composed slice domain. This avoids information loss and minimizes interference from unrelated objects.

**Adaptive Interaction Allocation** In a multi-solid system, the deformation of a deformable solid will be influenced by all loads and contact constraints in the system to varying extents. For example, among all loads, those that act directly on the deforming solid should have a more significant impact on its deformation than those far away from it. Therefore, we propose an adaptive interaction allocation mechanism to comprehensively take account all loads and contact constraints. Technically, the equivalent contact constraint $\bar{\mathbf{c}} \in \mathbb{R}^{M \times C}$ and the equivalent load $\bar{\mathbf{f}} \in \mathbb{R}^{M \times C}$ are formulated as follows:

$$\mathbf{c}_i' = \text{Linear}(\mathbf{c}_i), \quad \bar{\mathbf{c}} = \sum_{i=1}^{N^c} \frac{\mathbf{c}_i'}{\sum_{i=1}^{N^c} \mathbf{c}_i'} \mathbf{c}_i$$
$$\mathbf{f}_i' = \text{Linear}(\mathbf{f}_i), \quad \bar{\mathbf{f}} = \sum_{i=1}^{N^f} \frac{\mathbf{f}_i'}{\sum_{i=1}^{N^f} \mathbf{f}_i'} \mathbf{f}_i \tag{6}$$

The weights are learnable and deprived from corresponding physical interactions. Through the mechanism, those interactions have significant influence on the corresponding deformable solid having higher weights. At the same time, this allocation process can also be seen as the slice composition with prior inductive biases. In particular, the allocation weights for each deformable solid are different and are learned individually, extending the flexibility.

**Deformation Module** We adopt distinct deformation modules for each deformable solid. The input for $i$-th module is the deformation triplet $(\mathbf{d}_i, \bar{\mathbf{f}}, \bar{\mathbf{c}})$. Similarly, we employ attention mechanism and slice composition to model the deformation interaction as follows:

$$\mathbf{Q}, \mathbf{K}, \mathbf{V} = \text{Linear}(\mathbf{d}_i + \bar{\mathbf{f}} + \bar{\mathbf{c}})$$
$$\hat{\mathbf{d}}_i = \text{Softmax}\left(\frac{\mathbf{Q}\mathbf{K}^T}{\sqrt{C}}\right)\mathbf{V} \tag{7}$$

Here, $\hat{\mathbf{d}} = \{\hat{\mathbf{d}}_i \in \mathbb{R}^{M \times C}\}_{i=1}^{N^d}$ are updated tokens of deformable solids. Within the $i$-th deformation module, we explicitly establish the relationship between the $i$-th deformable solid and its associated equivalent load and contact

constraint, rather than implicitly learning the triplet relationship from data as in implicit modeling methods. This avoids the information confusion and ambiguity caused by the excessive number of solids. Besides, using an independent deformation module for each deformable solid allows for precise, individualized modeling of each solid, enhancing scalability and flexibility.

**Decoder** Afterwards, the transited tokens of deformable solids $\hat{\mathbf{d}} = \{\hat{\mathbf{d}}_i \in \mathbb{R}^{M \times C}\}_{i=1}^{N^d}$ are projected back to mesh points. Because these tokens are normalized during embedding, the decoding of mesh points and edges is naturally decoupled. We directly decode tokens with formulation proposed in (Wu et al., 2024):

$$\hat{\mathbf{u}}_i^d = \sum_{j=1}^{M} \mathbf{w}_{i,j} \hat{\mathbf{d}}_j, \quad 1 \leq i \leq N_i^d \tag{8}$$

Each token $\hat{\mathbf{d}}_j$ is projected back by weighted broadcast, where the weights are the same as those in the forward embedding in Eq.(1). The $\hat{\mathbf{u}}^d = \{\hat{\mathbf{u}}_i^d \in \mathbb{R}^{N_i^d \times C_d}\}_{i=1}^{N^d}$ are the prediction of the original deformable solids $\mathbf{u}^d$.

# 4. Experiments

We evaluate Unisoma on extensive experiments, including two essential tasks and seven well-established datasets, covering multi-solid systems with varying complexity.

## 4.1. Experiment Settings

**Datasets** Our experiments span different complexity from few solids to multiple solids. All of them are in 3D space. Deforming Plate (Pfaff et al., 2021), Cavity Grasping (Linkerhägner et al., 2023), Tissue Manipulation (Linkerhägner et al., 2023) and Rice Grip (Li et al., 2019) are public datasets, widely followed on autoregressive task. To explore more complex scenarios, we construct three datasets: Bilateral Stamping, Unilateral Stamping and Cavity Extruding, which are inspired by metal stamping (Wang & Budiansky, 1978) in industrial manufacturing and robotic gripping (Zhang et al., 2020). Except for the target geometry, we also predict attendant physical quantities in some experiments, including inner stress (Stress) and equivalent plastic strain (PEEQ). The summary of datasets is recorded in Table 1. See Appendix D for more details about datasets.

**Baselines** We comprehensively compare Unisoma against more than ten baselines within the implicit modeling paradigm. These include typical neural operators: GNO (Li et al., 2020b), Geo-FNO (Li et al., 2023b), GINO (Li et al., 2024a), LSM (Wu et al., 2023); Transformer solvers: Galerkin Transformer (Cao, 2021), Factformer (Li et al., 2024b), OFormer (Li et al., 2023c), ONO (Xiao et al., 2024), Transolver (Wu et al., 2024); and GNN-based mod-

*Table 1.* Summary of experiment datasets, which include various complexity. #Content records the number of deformable solids (DS), rigid solids (RS), loads (L) and contact pairs (C). #Mesh represents the mean number of discretized mesh points, including deformable solids and rigid solids.

| DATASET | #DIM | #MATERIAL | #CONTENT | #MESH | #TARGET |
|---|---|---|---|---|---|
| DEFORMING PLATE | 3D | HYPERELASTICITY | 1 DS 1 RS 1 L 1 C | 1271 | GEOMETRY STRESS |
| CAVITY GRASPING | 3D | ELASTICITY | 1 DS 2 RS 2 L 2 C | 1386 | GEOMETRY |
| TISSUE MANIPULATION | 3D | ELASTICITY | 1 DS 2 RS 2 L 2 C | 362 | GEOMETRY |
| RICE GRIP | 3D | ELASTO-PLASTICITY | 1 DS 2 RS 2 L 2 C | 1106 | GEOMETRY |
| BILATERAL STAMPING | 3D | ELASTO-PLASTICITY | 2 DS 2 RS 2 L 5 C | 13714 | GEOMETRY STRESS PEEQ |
| UNILATERAL STAMPING | 3D | ELASTO-PLASTICITY | 2 DS 17 RS 16 L 18 C | 49386 | GEOMETRY STRESS PEEQ |
| CAVITY EXTRUDING | 3D | ELASTO-PLASTICITY ELASTICITY | 3 DS 4 RS 4 L 6 C | 4800 | GEOMETRY STRESS PEEQ |

els: GraphSAGE (Hamilton et al., 2017), GraphUNet (Gao & Ji, 2019), MGN (Pfaff et al., 2021), HOOD (Grigorev et al., 2023), HCMT (Yu et al., 2024). These advanced deep models are widely used for continuum mechanics, and most have demonstrated success in few-solid scenarios.

**Implementation** For Unisoma, we set the number of processors, the hidden channels and the slice number to 2, 128 and 32, respectively, across all experiments. All experiments are conducted on a single RTX 3090 GPU and repeated three times. We utilize Relative L2 and Root Mean Square Error (RMSE) as evaluation metrics for long-time prediction and autoregressive simulation, respectively. See Appendix E and C for detailed implementations and metrics.

*Table 2.* Performance comparison on long-time prediction task. Relative L2 is recorded. A smaller value indicates better performance. The best result is in **bold** and the second best is underlined.

| | DEFORMING PLATE | | CAVITY GRASPING | TISSUE MANIPU-LATION |
|---|---|---|---|---|
| | GEOMETRY | STRESS | GEOMETRY | GEOMETRY |
| GEO-FNO | 0.0931 | 0.5048 | 0.1361 | 0.0764 |
| GINO | 0.1071 | 0.6015 | 0.1523 | 0.0781 |
| GNO | 0.1223 | 0.6362 | 0.1273 | 0.0769 |
| LSM | 0.1319 | 0.7002 | 0.1073 | 0.0399 |
| GALERKIN | 0.1368 | 0.7046 | 0.1529 | 0.0832 |
| FACTFORMER | 0.0945 | 0.5135 | 0.1085 | 0.0886 |
| OFORMER | 0.1091 | 0.6136 | 0.1112 | 0.0316 |
| ONO | 0.1269 | 0.6592 | 0.1191 | 0.0513 |
| TRANSOLVER | 0.0933 | 0.4968 | 0.1077 | 0.0363 |
| GRAPHSAGE | 0.1194 | 0.6160 | 0.1194 | 0.0794 |
| MGN | 0.1183 | 0.6232 | 0.1141 | 0.0825 |
| **UNISOMA** | **0.0892** | **0.4713** | **0.0984** | **0.0253** |

### 4.2. Experiment Results

**Long-Time Prediction** As presented in Table 2, we first conduct long-term prediction task on three public datasets,

commonly used in autoregressive settings (Pfaff et al., 2021; Linkerhägner et al., 2023). We directly predict the physical quantities for the step with the maximum applied load, using the first step. Unisoma achieves consistent state-of-the-art performance across these benchmarks. The deforming plate only involves two solids and one pair of contact. There is just one contact module and the adaptive interaction allocation module is omitted. Under these conditions, the advantage of explicit modeling is less pronounced, so the accuracy of advanced models remains similar. For tissue manipulation, one jaw of the gripper holds the tissue in a fixed position, while the other jaw moves. Because the physical interactions of these two states differ significantly, accurate modeling is more challenging. By explicitly modeling these contact constraints and loads, Unisoma achieves noticeable improvements in this benchmark.

We further evaluate more complex scenarios involving multiple solids. Metal stamping (Wang & Budiansky, 1978), one of the most crucial industrial processes, is examined in two settings (see Figure 4 and Appendix D): one featuring bilateral stamping by two rigid dies, and another involving multi-point unilateral stamping by sixteen rigid dies. The processed metal is hollow, and a rubber material is inserted to support its cross-section. Note that both scenarios are challenging, as they require the model to account for numerous solids and their interactions. As presented in Table 3, Unisoma also excels in complex scenarios when compared to implicit modeling. Beyond geometry, Unisoma delivers the best performance on stress and PEEQ, which are essential for subsequent design analysis. It is worth noting that Transolver outperforms other baselines in most targets, illustrating how slices effectively capture complex interactions. The performance margins with Unisoma highlight the strengths of explicit modeling in complicated setups. Moreover, in the bilateral stamping, random cuts in the metal can lead to occasional contact between the stamping dies and the rubber. Even if contact does not occur, we still

*Table 3.* Performance comparison on long-time prediction task. Relative L2 is recorded. A smaller value indicates better performance. The best result is in **bold** and the second best is underlined. Under the same target, the left column represents the results for the metal, while the right column represents the results for the rubber.

| | BILATERAL STAMPING | | | | | | UNILATERAL STAMPING | | | | | |
| --- | --- | --- | --- | --- | --- | --- | --- | --- | --- | --- | --- | --- |
| | GEOMETRY | | STRESS | | PEEQ | | GEOMETRY | | STRESS | | PEEQ | |
| GEO-FNO | 0.0125 | 0.0117 | 0.0390 | 0.1612 | 0.4269 | 0.4731 | 0.0109 | 0.0154 | 0.1113 | 0.1124 | 0.3771 | 0.3574 |
| GINO | 0.0299 | 0.0243 | 0.0543 | 0.2621 | 0.5523 | 0.6409 | 0.0188 | 0.0219 | 0.1262 | 0.1438 | 0.4651 | 0.4885 |
| GNO | 0.0285 | 0.0281 | 0.0463 | 0.2524 | 0.5830 | 0.5909 | 0.0426 | 0.0424 | 0.1351 | 0.1814 | 0.6413 | 0.6859 |
| LSM | 0.0598 | 0.0615 | 0.0556 | 0.3616 | 0.7001 | 0.7347 | 0.0367 | 0.0357 | 0.1327 | 0.1335 | 0.5889 | 0.6026 |
| GALERKIN | 0.0599 | 0.0594 | 0.0563 | 0.3442 | 0.6962 | 0.7335 | 0.1007 | 0.1010 | 0.1725 | 0.2794 | 0.7986 | 0.8555 |
| FACTFORMER | 0.0297 | 0.0105 | 0.0459 | 0.1378 | 0.6266 | 0.4218 | 0.0178 | 0.0166 | 0.1095 | 0.1035 | 0.3886 | 0.3022 |
| OFORMER | 0.0334 | 0.0268 | 0.0495 | 0.3178 | 0.5551 | 0.5157 | 0.0119 | 0.0098 | 0.1103 | 0.0981 | 0.3269 | 0.2675 |
| ONO | 0.0301 | 0.0275 | 0.0464 | 0.3291 | 0.5176 | 0.5231 | 0.0433 | 0.0437 | 0.1377 | 0.1311 | 0.5140 | 0.4748 |
| TRANSOLVER | 0.0083 | 0.0081 | 0.0346 | 0.1425 | 0.3592 | 0.3276 | 0.0079 | 0.0075 | 0.0839 | 0.0862 | 0.2938 | 0.2693 |
| GRAPHSAGE | 0.0227 | 0.0229 | 0.0440 | 0.2196 | 0.4936 | 0.4495 | 0.0275 | 0.0260 | 0.1059 | 0.1189 | 0.5441 | 0.5106 |
| MGN | 0.0195 | 0.0197 | 0.0421 | 0.2059 | 0.4646 | 0.4141 | 0.0391 | 0.0383 | 0.1252 | 0.1417 | 0.5805 | 0.5694 |
| **UNISOMA** | **0.0057** | **0.0052** | **0.0278** | **0.1039** | **0.2817** | **0.2265** | **0.0071** | **0.0063** | **0.0807** | **0.0773** | **0.2591** | **0.2114** |

*Table 4.* Performance comparison on autoregressive simulation task. We record RMSE-all, the average RMSE of the whole rollout trajectory and all samples. A smaller value indicates better performance.

| | CAVITY GRASPING ($\times 10^{-3}$) | TISSUE MANIPULATION ($\times 10^{-3}$) | RICE GRIP ($\times 10^{-3}$) | CAVITY EXTRUDING ($\times 10^{-2}$) | | |
| --- | --- | --- | --- | --- | --- | --- |
| GRAPHSAGE | $13.41 \pm 0.36$ | $13.19 \pm 0.31$ | $26.68 \pm 1.86$ | $65.81 \pm 1.58$ | $128.48 \pm 6.05$ | $104.44 \pm 4.79$ |
| GRAPHUNET | $14.27 \pm 0.11$ | $21.31 \pm 1.17$ | $22.48 \pm 1.23$ | $58.83 \pm 0.88$ | $96.26 \pm 1.44$ | $82.55 \pm 1.69$ |
| HOOD | $12.43 \pm 0.34$ | $9.87 \pm 0.17$ | $19.25 \pm 0.43$ | $53.98 \pm 1.84$ | $89.93 \pm 2.69$ | $81.02 \pm 2.68$ |
| HCMT | $19.32 \pm 1.82$ | $17.59 \pm 1.81$ | $18.36 \pm 0.47$ | $64.91 \pm 3.39$ | $110.18 \pm 5.98$ | $97.09 \pm 4.42$ |
| MGN | $12.89 \pm 0.46$ | $9.56 \pm 0.29$ | $19.61 \pm 0.78$ | $57.37 \pm 1.62$ | $95.87 \pm 1.17$ | $86.28 \pm 4.93$ |
| **UNISOMA** | $\mathbf{9.50 \pm 0.54}$ | $\mathbf{7.51 \pm 0.28}$ | $\mathbf{17.68 \pm 0.36}$ | $\mathbf{11.98 \pm 0.27}$ | $\mathbf{19.37 \pm 0.72}$ | $\mathbf{19.46 \pm 1.82}$ |

employ contact modules for potential interactions, and the adaptive interaction allocation mechanism controls the relevant constraints. This demonstrates both the scalability and flexibility of explicit modeling.

**Autoregressive Simulation** We further evaluate the performance of Unisoma on the autoregressive simulation task. In this task, the process starts from the first step, and each subsequent step is predicted based on the previously predicted step. This is more challenging because it requires accurate modeling to prevent physical shifts and error accumulation during the simulation. We compare Unisoma with advanced GNN-based simulators. As shown in Table 4, Unisoma achieves the best results across all datasets. In scenarios involving only one deformable solid, the physical interactions between the deformable solid and rigid solids are relatively straightforward to handle, resulting in similar performance between Unisoma and other GNN-based simulators. However, in scenarios with multiple solids, the advantage of explicit modeling becomes evident. For the cavity extrusion, the materials of the three deformable solids vary, and the number of contacts and loads is higher. Relying solely on message passing between nodes is insufficient to implicitly capture all physical interactions. Moreover, mes-

sage passing is constrained within a limited radius, which weakens the long-distant propagation of the influence of contacts. This limitation becomes more pronounced as the number of contacts increases. In contrast, Unisoma leverages explicit modeling to construct relationships between contacting solids and captures their influence on the deformation. The attention mechanism and slice composition enable global modeling, while the incorporation of mesh edges in the embedding preserves local features. As a result, the performance improvement in multi-solid scenarios is more significant. This modeling approach, which balances global and local considerations, also highlights the flexibility of explicit modeling.

**Out-of-distribution (OOD) generalization** To further examine the generalizability of Unisoma, we experiment with OOD long-time prediction task on the cavity extruding dataset. In this dataset, the Young's modulus and Poisson's ratio of the inner elastic material in the cavity are uniformly sampled from the ranges [30000, 70000] MPa and [0.1, 0.45]. The data is divided into two parts: the first part serves as the training and validation set, with parameter ranges of [30000, 65000] MPa and [0.1, 0.42]. The samples out of these scopes are used as the test set. The numbers

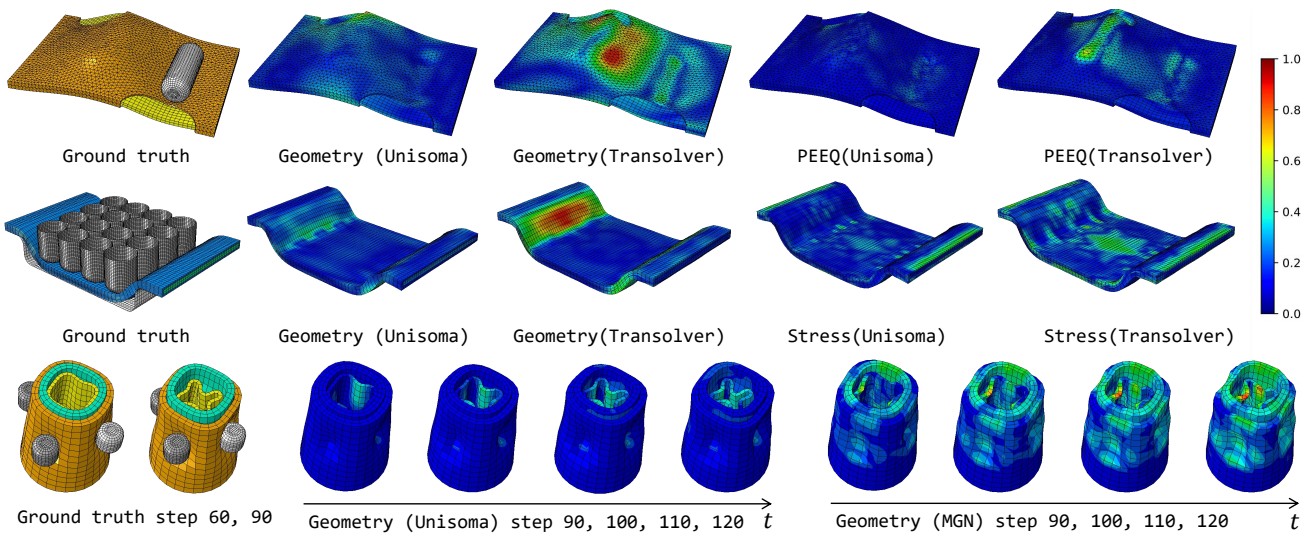

*Figure 4.* Visualization of error maps. The first, second, and third rows respectively show sample visualizations from the bilateral stamping, unilateral stamping, and cavity extruding datasets. The error has been normalized among the same physical quantities of the same samples.

*Table 5.* OOD generalization experiment on the cavity extruding dataset. Relative L2 is recorded. "Outer", "Middle", and "Inner" refer to the layers of cavity from the outside to the inside. See Appendix G for full results, where Unisoma still achieves best.

| | OUTER | MIDDLE | INNER | |
|---|---|---|---|---|
| | GEOMETRY | GEOMETRY | GEOMETRY | STRESS |
| GEO-FNO | 0.0117 | 0.0177 | 0.0198 | 0.2526 |
| GNO | 0.0238 | 0.0444 | 0.0449 | 0.3727 |
| LSM | 0.0249 | 0.0436 | 0.0339 | 0.3111 |
| FACTFORMER | 0.0202 | 0.0269 | 0.0308 | 0.2739 |
| OFORMER | 0.0134 | 0.0204 | 0.0229 | 0.2596 |
| ONO | 0.0213 | 0.0357 | 0.0372 | 0.2511 |
| TRANSOLVER | 0.0162 | 0.0334 | 0.0336 | 0.2643 |
| GRAPHSAGE | 0.0211 | 0.0339 | 0.0416 | 0.3571 |
| MGN | 0.0178 | 0.0280 | 0.0385 | 0.3526 |
| **UNISOMA** | **0.0077** | **0.0157** | **0.0179** | **0.2348** |

of samples in the training, validation and test set are 900, 83 and 217. As presented in Table 5, Unisoma can handle OOD samples well, where it consistently performs best on unseen materials. These results indicate that Unisoma also captures some generalizable physical interactions, further highlighting the advantage of explicit modeling.

**Efficiency** We provide the model efficiency comparison in Table 6. We can observe that the GPU memory usage of Unisoma is significantly lower than other models, including Transolver, which also uses slice. Even when the number of mesh points around 50,000 (unilateral stamping), the memory consumption still remains around 1GiB. Unisoma employs explicit modeling, processing each object separately. It breaks down the large matrix multiplications in implicit modeling into smaller, multiple matrix multipli-

*Table 6.* Efficiency comparison. The running time is measured by the time to complete one epoch, which contains $10^3$ iterations. The $k$ for Unisoma is set as 4 for efficiency test. Other edge-related methods set $k = 3$ or $4$ based on the memory usage. We record the max GPU memory in one epoch due to the dynamic mesh number.

| | BILATERAL STAMPING | | | UNILATERAL STAMPING | | |
|---|---|---|---|---|---|---|
| | PARAM (M) | TIME (S) | MEM (GIB) | PARAM (M) | TIME (S) | MEM (GIB) |
| GINO | 2.70 | 75.64 | 17.63 | 5.61 | 181.26 | 22.82 |
| GNO | 0.56 | 98.64 | 21.79 | 0.43 | 261.73 | 21.31 |
| OFORMER | 2.63 | 127.59 | 13.94 | 2.06 | 479.85 | 23.01 |
| ONO | 3.27 | 66.55 | 5.46 | 5.13 | 311.60 | 21.85 |
| TRANSOLVER | 3.81 | 83.97 | 6.35 | 5.07 | 402.35 | 16.33 |
| **UNISOMA** | 2.85 | 70.96 | 0.93 | 5.21 | 152.55 | 1.10 |

cations. This approach reduces the memory required for intermediate states during computation, allowing for more flexible memory allocation and release. As a result, the model can handle larger problem scales, making it far more applicable and meaningful for real-world applications.

## 5. Conclusions

This paper introduces a new paradigm of explicit modeling to solve complex multi-solid systems and, based on it, designing a unified and flexible Transformer architecture named Unisoma. Compared with implicit modeling models, Unisoma explicitly represents the key factors affecting solid deformation and employs deformation triplet to more accurately capture the diverse interactions among multiple solids. Extensive experiments are provided to verify the performance, OOD generalizability and efficiency.

## Acknowledgements

This work is supported partly by National Key Reasearch Plan under grant No.2024YFC2607404, the National Natural Science Foundation of China (NSFC) 61925202, and the Jiangsu Provincial Key Research and Development Program under Grant BE2022065-1, BE2022065-3.

## Impact Statement

This paper presents work whose goal is to advance the field of Machine Learning for Multi-solid Systems. Note that this work mainly focuses on the scientific computation. We are entirely committed to ensuring ethical considerations are taken into account when doing our research. There are many potential societal consequences of our work, none which we feel must be specifically highlighted here.

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

# A. Overall Structure

The overall structure of Unisoma is as follows:

**Encoder** Given a system with multiple objects composed of $N^d$ deformable solids $\mathbf{u}^d = \{\mathbf{u}_i^d \in \mathbb{R}^{N_i^d \times C_d}\}_{i=1}^{N^d}$ with different material properties, $N^r$ rigid solids $\mathbf{u}^r = \{\mathbf{u}_i^r \in \mathbb{R}^{N_i^r \times C_r}\}_{i=1}^{N^r}$, and $N^f$ loads $\mathbf{u}^f = \{\mathbf{u}_i^f \in \mathbb{R}^{N_i^f \times C_f}\}_{i=1}^{N^f}$, we first embed each of them into edge augmented physics-aware tokens by Eq.(1):

$$
\begin{aligned}
\mathbf{x}_i^d &= \text{Linear}(\mathbf{u}_i^d), \quad \mathbf{d}_i = \text{Encoder}(\mathbf{x}_i^d), \quad 1 \le i \le N^d \\
\mathbf{x}_i^r &= \text{Linear}(\mathbf{u}_i^r), \quad \mathbf{r}_i = \text{Encoder}(\mathbf{x}_i^r), \quad 1 \le i \le N^r \\
\mathbf{x}_i^f &= \text{Linear}(\mathbf{u}_i^f), \quad \mathbf{f}_i = \text{Encoder}(\mathbf{x}_i^f), \quad 1 \le i \le N^f
\end{aligned}
$$

Here, $\mathbf{d} = \{\mathbf{d}_i \in \mathbb{R}^{M \times C}\}_{i=1}^{N^d}$, $\mathbf{r} = \{\mathbf{r}_i \in \mathbb{R}^{M \times C}\}_{i=1}^{N^r}$, and $\mathbf{f} = \{\mathbf{f}_i \in \mathbb{R}^{M \times C}\}_{i=1}^{N^f}$ are edge augmented physics-aware tokens of corresponding objects in $\mathbf{u}^d$, $\mathbf{u}^r$, and $\mathbf{u}^f$, respectively.

**Processor** We first characterize the contact constraints in the system. For two solids that may contact with each other with high probabilities, we utilize contact module to capture the contact constraint. For any two solids $\mathbf{g}_i$ and $\mathbf{g}_j$ that are likely to contact, where $\mathbf{g}_i, \mathbf{g}_j \in \mathbf{d} \cup \mathbf{r}$ and $1 \le i, j \le N^d + N^r$, the contact constraint $\mathbf{c}_k$ between them is formalized as:

$$
\mathbf{Q}, \mathbf{K}, \mathbf{V} = \text{Linear}(\mathbf{g}_i + \mathbf{g}_j)
$$

$$
\mathbf{c}_k = \text{Softmax}\left(\frac{\mathbf{Q}\mathbf{K}^T}{\sqrt{C}}\right)\mathbf{V}
$$

Here, totally $N^c$ contact modules are arranged to learn $N^c$ pairs of contact constraints $\mathbf{c} = \{\mathbf{c}_k \in \mathbb{R}^{M \times C}\}_{k=1}^{N^c}$.

Then we deploy adaptive interaction allocation mechanism to integrate loads and contact constraints, as follows:

$$
\mathbf{c}_i' = \text{Linear}(\mathbf{c}_i), \quad \hat{\mathbf{c}} = \sum_{i=1}^{N^c} \frac{\mathbf{c}_i'}{\sum_{i=1}^{N^c} \mathbf{c}_i'} \mathbf{c}_i, \quad \bar{\mathbf{c}} = \text{FFN}(\hat{\mathbf{c}}) + \hat{\mathbf{c}}
$$

$$
\mathbf{f}_i' = \text{Linear}(\mathbf{f}_i), \quad \bar{\mathbf{f}} = \sum_{i=1}^{N^f} \frac{\mathbf{f}_i'}{\sum_{i=1}^{N^f} \mathbf{f}_i'} \mathbf{f}_i
$$

The $\bar{\mathbf{c}}$ and $\bar{\mathbf{f}}$ are the equivalent contact constraint and equivalent load. There is little difference in their operation. As described above, the contact module is attention-based. When the number of contacts increases, the parameters grow and make the model unstable. Therefore, we move the FFN($\cdot$) to the back of contact constraints allocation, instead of arranging distinct FFN($\cdot$) for each contact module, to reduce parameters. With regard to loads, they are from encoders. The FFN($\cdot$) is unnecessary for them.

Subsequently, we adopt deformation modules to capture the deformation of deformable solids. Notably, each deformation module processes one deformable solid, and the equivalent load and contact constraint for each deformable solid are different. For $i$-th deformable solid, its deformation is modeled as:

$$
\mathbf{Q}, \mathbf{K}, \mathbf{V} = \text{Linear}(\mathbf{d}_i + \bar{\mathbf{f}} + \bar{\mathbf{c}})
$$

$$
\mathbf{d}_i' = \text{Softmax}\left(\frac{\mathbf{Q}\mathbf{K}^T}{\sqrt{C}}\right)\mathbf{V}
$$

$$
\hat{\mathbf{d}} = \mathbf{d}_i' + \text{FFN}(\mathbf{d}_i')
$$

Here, $\hat{\mathbf{d}} = \{\hat{\mathbf{d}}_i \in \mathbb{R}^{M \times C}\}_{i=1}^{N^d}$ are updated tokens of deformable solids.

**Decoder** We decode updated tokens back to mesh points with formulation proposed in (Wu et al., 2024):

$$
\hat{\mathbf{u}}_i^d = \text{FFN}(\text{Decoder}(\hat{\mathbf{d}}_j) + \mathbf{x}_i^d) = \text{FFN}(\sum_{j=1}^{M} \mathbf{w}_{i,j}\hat{\mathbf{d}}_j + \mathbf{x}_i^d), \quad 1 \le i \le N_i^d
$$

Each token $\hat{\mathbf{d}}_j$ is projected back to mesh points by weighted broadcast, where the weights are the same as those in the forward embedding in Eq.(1). The $\hat{\mathbf{u}}^d = \{\hat{\mathbf{u}}_i^d \in \mathbb{R}^{N_i^d \times C_d}\}_{i=1}^{N^d}$ are the predicted states corresponding to the original input deformable solids $\mathbf{u}^d$. The residual connection with $\mathbf{x}^d$ is introduced to prevent the vanishing gradient problem.

## B. Slice Composition and Decomposition

For deep features $\mathbf{x}^\alpha \in \mathbb{R}^{N^\alpha \times C}$ and $\mathbf{x}^\beta \in \mathbb{R}^{N^\beta \times C}$ of any two objects, we embed them into slice $\mathbf{z}^{\alpha\beta} \in \mathbb{R}^{M \times C}$ as follows:

$$\mathbf{z}_j^{\alpha\beta} = \frac{\sum_{i=1}^{N^\alpha + N^\beta} \mathbf{w}_{i,j}^{\alpha\beta} \text{Contact}(\mathbf{x}^\alpha, \mathbf{x}^\beta)_i + \gamma \sum_{\mathbf{e}_{p,q}^{\alpha\beta} \in E^\alpha \cup E^\beta} \mathbf{w}_{(p,q),j}^{e^{\alpha\beta}} \mathbf{e}_{p,q}^{\alpha\beta}}{\sum_{i=1}^{N^\alpha + N^\beta} \mathbf{w}_{i,j}^{\alpha\beta} + \gamma \sum_{\mathbf{e}_{p,q}^{\alpha\beta} \in E^\alpha \cup E^\beta} \mathbf{w}_{(p,q),j}^{e^{\alpha\beta}}} \tag{9}$$

Here, $\mathbf{e}_{p,q}^\alpha \in E^\alpha$ and $\mathbf{e}_{p,q}^\beta \in E^\beta$ are corresponding edge sets. Since we apply $k$-NN separately to different solids, using the same value of $k$ for each, it follows that $E^\alpha \cap E^\beta = \varnothing$ and $\gamma = k$. Therefore, similar to $\mathbf{w}^{\alpha\beta}$, the weight $\mathbf{w}^{e^{\alpha\beta}}$ can also be separated according to $\alpha$ and $\beta$, allowing the Eq.(9) to be rewritten as:

$$\mathbf{z}_j^{\alpha\beta} = \frac{\sum_{i=1}^{N^\alpha} \mathbf{w}_{i,j}^\alpha \mathbf{x}_i^\alpha + \gamma \sum_{\mathbf{e}_{p,q}^\alpha \in E^\alpha} \mathbf{w}_{(p,q),j}^{e^\alpha} \mathbf{e}_{p,q}^\alpha + \sum_{i=1}^{N^\beta} \mathbf{w}_{i,j}^\beta \mathbf{x}_i^\beta + \gamma \sum_{\mathbf{e}_{p,q}^\beta \in E^\beta} \mathbf{w}_{(p,q),j}^{e^\beta} \mathbf{e}_{p,q}^\beta}{\sum_{i=1}^{N^\alpha} \mathbf{w}_{i,j}^\alpha + \gamma \sum_{\mathbf{e}_{p,q}^\alpha \in E^\alpha} \mathbf{w}_{(p,q),j}^{e^\alpha} + \sum_{i=1}^{N^\beta} \mathbf{w}_{i,j}^\beta + \gamma \sum_{\mathbf{e}_{p,q}^\beta \in E^\beta} \mathbf{w}_{(p,q),j}^{e^\beta}} \tag{10}$$

As mentioned in Section 3.2, the edge weights $\mathbf{w}^{e^\beta}$ is deprived from point weights $\mathbf{w}^\beta$. When $\mathbf{w}^\beta = \mathbf{0}$, we have $\mathbf{w}^{e^\beta} = \mathbf{0}$. Following this formulation, embedding each object individually can be formulated as:

$$\begin{aligned}
\mathbf{z}_j^\alpha &= \frac{\sum_{i=1}^{N^\alpha} \mathbf{w}_{i,j}^\alpha \mathbf{x}_i^\alpha + \gamma \sum_{\mathbf{e}_{p,q}^\alpha \in E^\alpha} \mathbf{w}_{(p,q),j}^{e^\alpha} \mathbf{e}_{p,q}^\alpha + \sum_{i=1}^{N^\beta} \mathbf{w}_{i,j}^\beta \mathbf{x}_i^\beta + \gamma \sum_{\mathbf{e}_{p,q}^\beta \in E^\beta} \mathbf{w}_{(p,q),j}^{e^\beta} \mathbf{e}_{p,q}^\beta}{\sum_{i=1}^{N^\alpha} \mathbf{w}_{i,j}^\alpha + \gamma \sum_{\mathbf{e}_{p,q}^\alpha \in E^\alpha} \mathbf{w}_{(p,q),j}^{e^\alpha} + \sum_{i=1}^{N^\beta} \mathbf{w}_{i,j}^\beta + \gamma \sum_{\mathbf{e}_{p,q}^\beta \in E^\beta} \mathbf{w}_{(p,q),j}^{e^\beta}} \quad (\mathbf{w}^\beta = \mathbf{w}^{e^\beta} = \mathbf{0}) \\
&= \frac{\sum_{i=1}^{N^\alpha} \mathbf{w}_{i,j}^\alpha \mathbf{x}_i^\alpha + \gamma \sum_{\mathbf{e}_{p,q}^\alpha \in E^\alpha} \mathbf{w}_{(p,q),j}^{e^\alpha} \mathbf{e}_{p,q}^\alpha}{\sum_{i=1}^{N^\alpha} \mathbf{w}_{i,j}^\alpha + \gamma \sum_{\mathbf{e}_{p,q}^\alpha \in E^\alpha} \mathbf{w}_{(p,q),j}^{e^\alpha}}
\end{aligned} \tag{11}$$

Therefore, we can also conduct slice decomposition with mesh edges, which builds a pure slice domain on the holistic input domain but only projected by a single object.

Furthermore, the slice $\mathbf{z}_j^{\alpha\beta}$ can be reformulated as follows:

$$\begin{aligned}
\mathbf{z}_j^{\alpha\beta} &= \frac{\sum_{i=1}^{N^\alpha} \mathbf{w}_{i,j}^\alpha \mathbf{x}_i^\alpha + \gamma \sum_{\mathbf{e}_{p,q}^\alpha \in E^\alpha} \mathbf{w}_{(p,q),j}^{e^\alpha} \mathbf{e}_{p,q}^\alpha + \sum_{i=1}^{N^\beta} \mathbf{w}_{i,j}^\beta \mathbf{x}_i^\beta + \gamma \sum_{\mathbf{e}_{p,q}^\beta \in E^\beta} \mathbf{w}_{(p,q),j}^{e^\beta} \mathbf{e}_{p,q}^\beta}{\sum_{i=1}^{N^\alpha} \mathbf{w}_{i,j}^\alpha + \gamma \sum_{\mathbf{e}_{p,q}^\alpha \in E^\alpha} \mathbf{w}_{(p,q),j}^{e^\alpha} + \sum_{i=1}^{N^\beta} \mathbf{w}_{i,j}^\beta + \gamma \sum_{\mathbf{e}_{p,q}^\beta \in E^\beta} \mathbf{w}_{(p,q),j}^{e^\beta}} \\
&= \frac{(\sum_{i=1}^{N^\alpha} \mathbf{w}_{i,j}^\alpha + \gamma \sum_{\mathbf{e}_{p,q}^\alpha \in E^\alpha} \mathbf{w}_{(p,q),j}^{e^\alpha}) \mathbf{z}_j^\alpha + (\sum_{i=1}^{N^\beta} \mathbf{w}_{i,j}^\beta + \gamma \sum_{\mathbf{e}_{p,q}^\beta \in E^\beta} \mathbf{w}_{(p,q),j}^{e^\beta}) \mathbf{z}_j^\beta}{\sum_{i=1}^{N^\alpha} \mathbf{w}_{i,j}^\alpha + \gamma \sum_{\mathbf{e}_{p,q}^\alpha \in E^\alpha} \mathbf{w}_{(p,q),j}^{e^\alpha} + \sum_{i=1}^{N^\beta} \mathbf{w}_{i,j}^\beta + \gamma \sum_{\mathbf{e}_{p,q}^\beta \in E^\beta} \mathbf{w}_{(p,q),j}^{e^\beta}} \\
&\approx \theta \mathbf{z}_j^\alpha + (1-\theta) \mathbf{z}_j^\beta
\end{aligned} \tag{12}$$

Here, with the mesh edges, the $\mathbf{z}^{\alpha\beta}$ can also be seen as the composition of $\mathbf{z}^\alpha$ and $\mathbf{z}^\beta$ through coefficient $\theta$, referred to as slice composition. The unified forms of Eq.(11), Eq.(12), Eq.(4) and Eq.(5) benefit from the unified aggregation of mesh edges in Eq.(1).

## C. Metrics

We employ different metrics for specific tasks, adhering to the evaluation approaches used in related works.

**Long-time Prediction: Relative L2** In line with prior studies on long-time prediction tasks (Raissi et al., 2019; Li et al., 2020a; Wu et al., 2024), we use the relative L2 to assess performance. Given the input physical quantities $\mathbf{u}$ and the predictions $\hat{\mathbf{u}}$, the relative L2 is computed as:

$$\text{Relative L2} = \frac{\|\mathbf{u} - \hat{\mathbf{u}}\|}{\|\mathbf{u}\|}$$

**Autoregressive Simulation: RMSE** Consistent with works focused on autoregressive simulation tasks (Pfaff et al., 2021; Yu et al., 2024; Li et al., 2019), we use Root Mean Square Error (RMSE) as the evaluation metric. Given the input physical quantities $\mathbf{u}$ and the predictions $\hat{\mathbf{u}}$, RMSE is calculated as:

$$\text{RMSE} = \sqrt{\frac{1}{N} \sum_{i=1}^{N} \|\mathbf{u}_i - \hat{\mathbf{u}}_i\|^2}$$

# D. Datasets

We extensively evaluate our model on two key tasks across seven datasets. These datasets encompass varying levels of complexity in the number of solids, diversity in solid materials, and variety in applied loads. The summary of datasets is recorded in Table 1.

## D.1. Public Datasets

**Deforming Plate** (Pfaff et al., 2021) This dataset consists of a 3D dynamic simulation of a deformable plate being pressed by a rigid solid (Figure 5), with a total of 1,200 samples. The deformable plate is made of a hyperelastic material, and the target physical quantities include the geometry and inner stress of the plate. Typically used for autoregressive tasks, this dataset spans 400 time steps. In our study, we evaluate the long-term prediction performance of Unisoma. The first time step is used as input, and we predict the results for the time step corresponding to the largest rigid solid movement (approximately step 340). Each sample contains an average of 1,271 points. Following the strategy outlined in the original paper, we use 1,000 samples for training, 100 for validation, and 100 for testing.

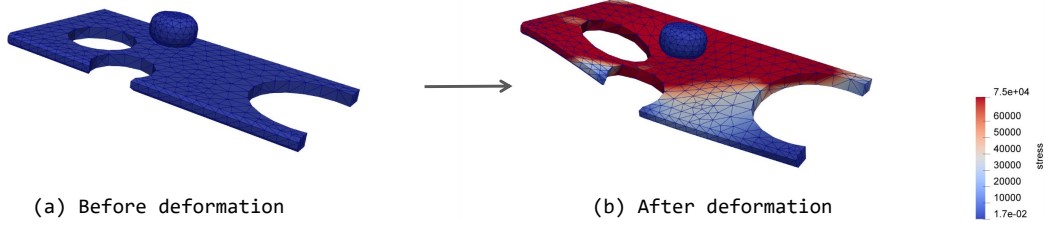

(a) Before deformation                    (b) After deformation

*Figure 5.* The deforming plate scenario.

**Cavity Grasping** (Linkerhägner et al., 2023) This dataset contains a 3D dynamic simulation of a deformable cavity grasped by a rigid gripper (Figure 6), with a total of 840 samples. The rigid gripper is treated as two rigid solids, as its two jaws move in different directions. The deformable cone-shaped cavities are randomly generated with radii ranging from 87.5 to 50. The materials of the cavities are elastic, with Poisson's ratios cyclically assigned from {-0.9, 0.0, 0.49}. Typically used for autoregressive tasks, this dataset spans 105 time steps. In our work, we evaluate both long-term prediction and autoregressive simulation tasks. For the long-term prediction task, we use the first time step as input and predict the outcome of the time step corresponding to the largest rigid solid movement (the last step). Each sample contains 1,386 points. Following the strategy in the original paper, 600 samples are used for training, 120 for validation, and 120 for testing.

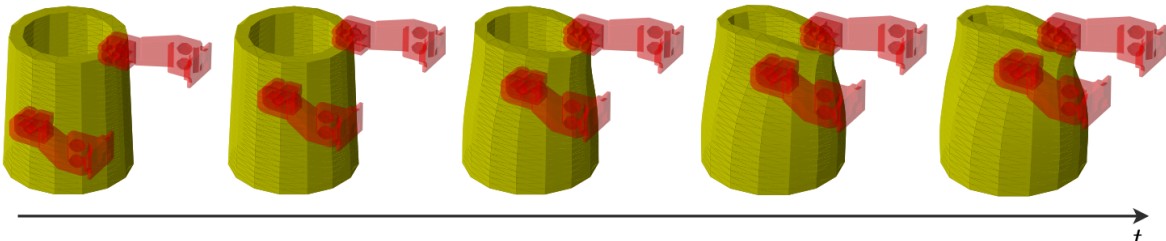

*Figure 6.* The cavity grasping scenario.

**Tissue Manipulation** (Linkerhägner et al., 2023) This dataset simulates the 3D dynamics of tissue deformation caused by a rigid gripper, a scenario often encountered in robot-assisted surgery. It contains 840 samples in total. The rigid gripper is treated as two rigid solids, as its two jaws exhibit distinct movement patterns. The tissue material is elastic, with Young's

modulus sampled from {10000, 80000, 30000}. This dataset spans 105 time steps and is commonly used for autoregressive tasks. In our work, we evaluate both long-term prediction and autoregressive simulation tasks. For long-term prediction, the first time step is used as input, and the outcome of the step with the largest rigid solid movement (the last step) is predicted. Each sample contains 363 points. Following the original paper, we allocate 600 samples for training, 120 for validation, and 120 for testing.

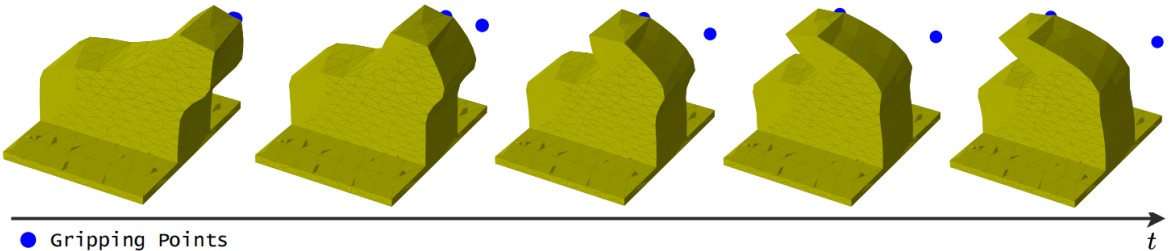

Figure 7. The tissue manipulation scenario.

**Rice Grip** (Li et al., 2019) This dataset involves the 3D dynamic simulation of sticky rice being gripped by two rigid grippers (Figure 8). The rice is modeled as an elasto-plastic material. Two parallel grippers, represented as cuboids, are initialized at random positions and orientations. We discretize each of them into 180 mesh points. During each trajectory, the grippers move closer together before returning to their original positions. The task involves learning the physical interactions between the grippers and the rice, as well as the deformation within the rice. The dataset spans 41 time steps and is typically used for autoregressive tasks. In our experiments, we evaluate the autoregressive simulation task on it. Each sample contains an average of 1,271 points. As for the experiment, we use 1200 samples in total and the numbers of samples used for training, validation and test are 1000, 100 and 100, respectively.

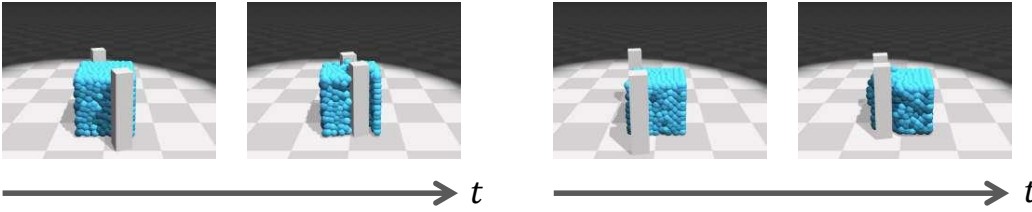

Figure 8. The rice grip scenario.

### D.2. Created Datasets

All public datasets mentioned before involve few solids and simple dynamics. To diversify the datasets and improve the complexity, we curate three multi-solid datasets with industrial inspiration for evaluation. These datasets are calculated by ABAQUS software (Abaqus, 2011).

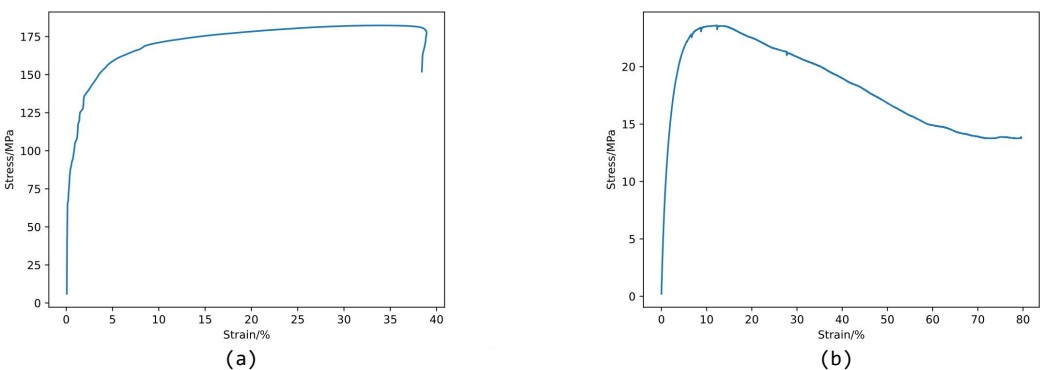

Figure 9. (a) the engineering stress-strain curve of the aluminum alloy; (b) The engineering stress-strain curve of the polyethylene rubber.

**Bilateral Stamping** Inspired by the metal stamping technique, we create a dataset related to bilateral stamping. As illustrated in Figure 10, this scenario involves stamping a hollow metal workpiece using two rigid stamping dies to form the desired target shapes. To maintain the integrity of the metal's cross-sectional shape during the process, a rubber insert

is squeezed into the hollow space while stamping. Both the metal and the rubber are modeled as elasto-plastic materials, with their material parameters derived from real aluminum alloy and polyethylene rubber. The Poisson's ratios for these materials are 0.37 and 0.27, while their Young's moduli are 69,000 MPa and 1,306.26 MPa, respectively. Their engineering stress-strain curves are depicted in Figure 9. In this dataset, two filleted cylinders serve as the stamping dies, and their positions and movement distances are randomly generated. The shape parameters of the solids are shown in Figure 11. All sample parameters are uniformly sampled from predefined intervals to ensure dataset diversity. This dataset is designed for the long-time prediction task, collecting the system's initial and final states. The target physical quantities include the geometry, inner stress, and plastic equivalent strain (PEEQ) of both the metal and the rubber. On average, each sample consists of 13,714 points. We generate a total of 1,200 samples, of which 1,000 are used for training, 100 for validation, and 100 for testing.

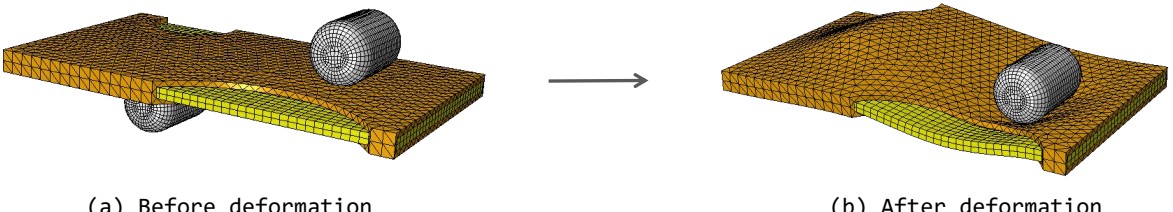

(a) Before deformation         (b) After deformation

*Figure 10.* The bilateral stamping scenario.

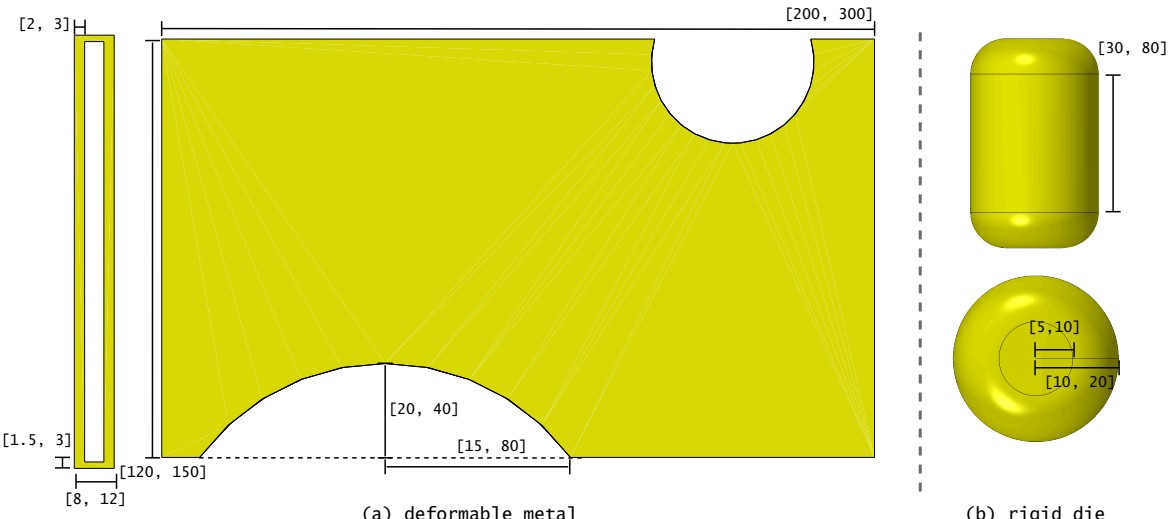

(a) deformable metal         (b) rigid die

*Figure 11.* The shape parameters (unit: $mm$) of the deformable metal and rigid dies in bilateral stamping dataset. The rubber shape is deprived from the metal.

**Unilateral Stamping** Building on the concept of metal stamping, we develop a dataset focused on multi-point unilateral stamping. As shown in Figure 12, this scenario simulates the deformation of a hollow metal workpiece using multiple rigid stamping dies—16 dynamic dies and 1 static die in this dataset—to achieve the target shapes. To preserve the cross-sectional integrity of the hollow metal during stamping, a rubber insert is pressed into the cavity. The material properties of the metal and rubber are consistent with those used in the bilateral stamping dataset. The shape parameters of the solids are detailed in Figure 13, with all sample parameters uniformly sampled within specified ranges to ensure diversity. This dataset is tailored for long-time prediction tasks, capturing both the initial and final states of the system. The target physical quantities include the geometry, internal stress, and plastic equivalent strain (PEEQ) of the metal and rubber. Each sample comprises an average of 49,386 points. A total of 1,200 samples are generated, distributed as 1,000 for training, 100 for validation, and 100 for testing.

**Cavity Extruding** Inspired by the robotic gripping (Zhang et al., 2020), we enhance the complexity of the cavity grasping dataset (Linkerhägner et al., 2023). As illustrated in Figure 14, this scenario simulates the deformation of multi-layer cavity using four rigid grippers. The grippers' positions and movement distances are randomly generated. The cavity is composed of three deformable layers: the outer and middle layers are elasto-plastic materials, while the inner layer is an elastic material.

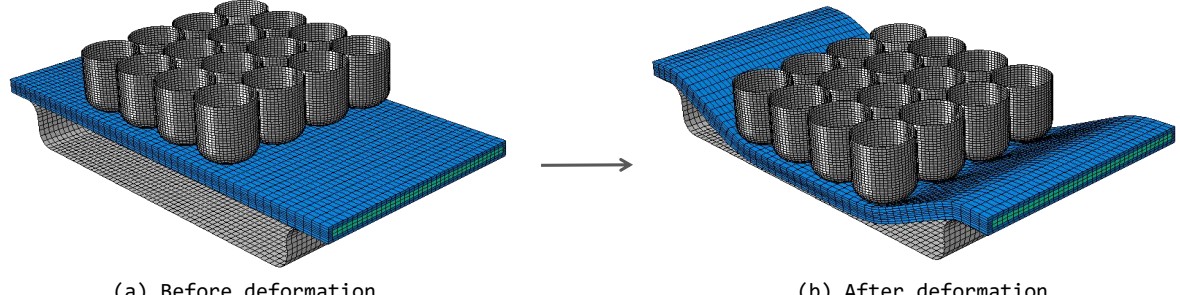

(a) Before deformation

(b) After deformation

*Figure 12.* The unilateral stamping scenario.

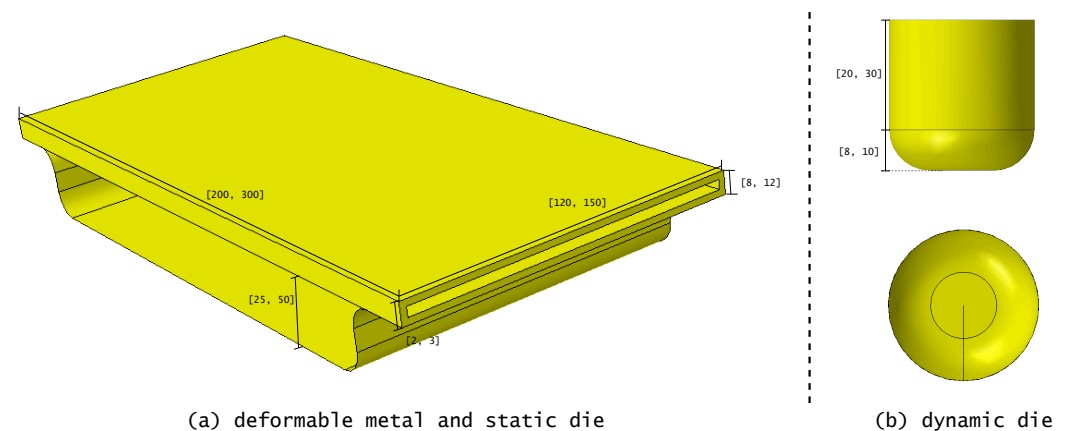

(a) deformable metal and static die

(b) dynamic die

*Figure 13.* The shape parameters (unit: $mm$) of the deformable metal and rigid dies in unilateral stamping dataset. The rubber shape is deprived from the metal.

The stress-strain curves for the elasto-plastic layers follow those shown in Figure 9, with the Young's modulus uniformly sampled from [65000, 75000] MPa and [35000, 45000] MPa, and the Poisson's ratio uniformly sampled from [0.3, 0.45] and [0.3, 0.45]. For the elastic innermost layer, the Young's modulus is uniformly sampled from [30000, 70000], and the Poisson's ratio is uniformly sampled from [0.1, 0.45]. This significantly increases the complexity of the data distribution. The shape parameters of the solids are provided in Figure 15, and all sample parameters are uniformly sampled within specified intervals to ensure dataset diversity. This dataset is designed for autoregressive simulation tasks and includes 121 steps from the initial state to the final state. For each step, the target physical quantities include the geometry, inner stress, and plastic equivalent strain (PEEQ) of the two elasto-plastic layers, as well as the geometry and inner stress of the elastic layer. Each sample contains 4,800 points, which is significantly larger than existing autoregressive simulation datasets. In total, we generated 1,200 samples, of which 1,000 are used for training, 100 for validation, and 100 for testing.

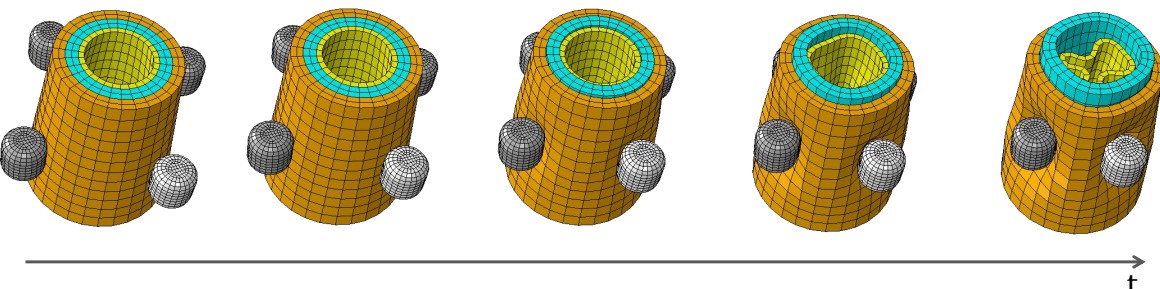

*Figure 14.* The cavity extruding scenario.

## E. Implementation Details

**Implementations**  As shown in Table 7, Unisoma and all baseline models are trained and tested using the same training strategy. We utilize relative L2 as loss function for long-time prediction task and MSE (Mean Square Error) for autoregressive

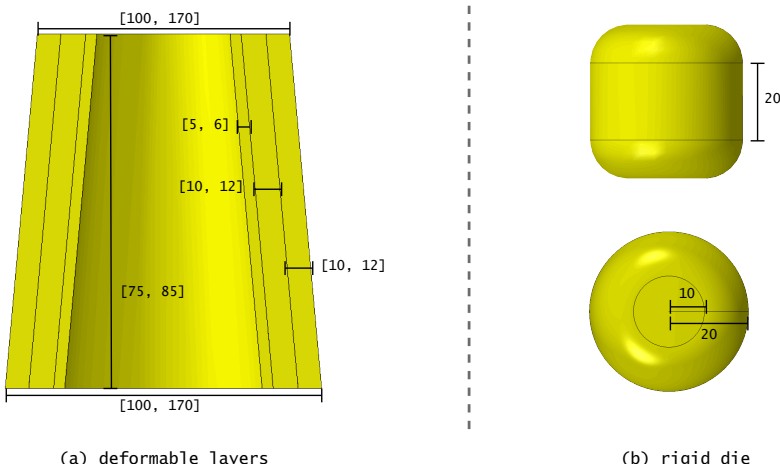

(a) deformable layers                    (b) rigid die

*Figure 15.* The shape parameters (unit: $mm$) of the deformable layers and rigid dies in cavity extruding dataset.

*Table 7.* Training and Model Configurations of Unisoma. The definition of batch size differs between the two tasks. For long-time prediction, the batch size refers to the number of samples in a batch. For autoregressive simulation, only one sample is processed during each forward and backward pass, and the batch size corresponds to the number of time steps in the sample. Since GPU memory usage varies across different models, the batch sizes of baseline models in the autoregressive simulation task are dynamically adjusted to avoid GPU memory overflow while maintaining performance.

| TASKS | DATASETS | TRAINING CONFIGURATION | | | | MODEL CONFIGURATION | | |
|---|---|---|---|---|---|---|---|---|
| | | LOSS | EPOCHS | LR | BATCH | PROCESSORS | CHANNELS | SLICES |
| LONG-TIME PREDICTION | DEFORMING PLATE CAVITY GRASPING TISSUE MANIPULATION | RELATIVE L2 | 200 | $3 \times 10^{-4}$ | 1 20 20 | 2 | 128 | 32 |
| | BILATERAL STAMPING UNILATERAL STAMPING | RELATIVE L2 | 100 | $3 \times 10^{-4}$ | 1 1 | | | |
| AUTOREGRESSIVE SIMULATION | CAVITY GRASPING TISSUE MANIPULATION RICE GRIP CAVITY EXTRUDING | MSE | 100 | $3 \times 10^{-4}$ | 104 104 40 120 | | | |

simulation task. For multiple physical quantities, we add each loss with equal weights. For Unisoma, we set the number of processors, hidden feature channels, and slices to 2, 128, and 32, respectively, across all experiments. To ensure fair comparisons, we first approximate the parameter count of all baseline models to match that of Unisoma and then adjust their parameters to minimize overfitting and achieve better performance. All experiments are conducted on a single RTX 3090 GPU (24GB memory) and repeated three times. Additionally, when adjusting the parameters of the baselines, we take maximum GPU memory usage into account as a constraint, which means the GPU memory usage of all models cannot incur "Out of memory" error. This is important for practical applications. Due to the number of modules is adaptable to the number of solids and their contacts, for all experiments, Unisoma performs well under a single set of network parameters, which demonstrates its robustness and generalizability across various tasks and datasets. This consistency eliminates the need for extensive parameter tuning when switching between tasks, making it more efficient and user-friendly for practical applications. Additionally, the ability of Unisoma to handle diverse scenarios with a unified configuration highlights its capacity to effectively model complex physical systems, where different dynamics and interactions are involved. This advantage becomes particularly significant when compared to baseline models, which often require task-specific parameter adjustments to achieve optimal performance.

Since these neural operators and transformer-based baselines employ implicit modeling and are primarily designed for Eulerian settings, they are not naturally suited for handling Lagrangian scenarios, such as multi-solid systems. Consequently, we preprocess the data to adapt it for use with these baselines. For different objects within the system, we first align the feature dimensions using padding, then concatenate along the length dimension to combine all objects into a single large Lagrangian sequence of points. For models that struggle to handle the Lagrangian setting (Li et al., 2023b; Wu et al.,

2023; Li et al., 2024b; 2023c), we map each point to a regular grid, transforming the data into an Eulerian representation. Specifically, for a point sequence of size $N \times C$, we discretize a cubic space $[x, y, z] \in [-1, 1]^3$ into a regular grid, with the number of discretization points along each axis set to $\lceil \sqrt[3]{N} \rceil$. We then map each point's coordinates and physical quantities to the corresponding grid points. Excess grid cells are padded to align dimensions. Experiments show that this approach improves baseline performance to some extent but results in a partial loss of information from the Lagrangian perspective, revealing limitations in addressing multi-solid problems effectively.

We implement baselines based on official and popular implementations. For autoregressive simulation task, we uniformly add noise with a mean of 0 and a variance of 0.001 to improve the error accumulation control during rollout. Because the results of MGN (Pfaff et al., 2021) on tissue manipulation and cavity grasping datasets are well-explored in GGNS (Linkerhägner et al., 2023), we directly use the model and weights in their code repository. Additionally, the data preprocessing code for pooling is not provided in HCMT (Yu et al., 2024). Therefore, we attempted to implement our own version, which, however, encountered convergence issues in some scenarios. We provide key network hyperparameters and parameter numbers of baselines in Table 10. We refer the readers to original papers and code repositories for details.

**Efficiency** In this paper, we report GPU memory usage based on the operating system's measurements, which include any memory pre-allocated by the PyTorch caching allocator. This approach yields a conservative yet realistic view of hardware requirements during training, since once memory is reserved, it is effectively unavailable for other processes. Even though the model may not be actively using all of the allocated space at each moment, including pre-allocation in the reported usage aligns more closely with the practical resource constraints encountered in real-world deployments. Furthermore, all models are evaluated under the default PyTorch memory usage strategy on a single RTX 3090 GPU, ensuring a fair comparison across different architectures and methods.

## F. Ablation Study

We include ablation studies in Table 8. In general, we observe that incorporating mesh edges benefits the final performance by enabling the model to capture more local features. However, as the number of mesh edges increases (especially when $k = 8$), the performance declines. This is because an excessive number of edges introduces redundant information and additional noise, which negatively impacts the model's effectiveness. Furthermore, increasing the number of edges significantly raises the computational cost, making the model less efficient. In principle, the optimal number of edges depends on the problem's scale and the number of mesh points. In our experiments, $k$ is easy-to-tune in the range of 3 to 5.

Besides, removing the loads (which are usually implicitly considered in existing models) will damage the model performance seriously, especially when the loads are more complex. This result further demonstrates the advantages of the explicit modeling in handling complex multi-solid systems.

*Table 8.* Ablation results. We experiment on two variants: the mesh edges and loads. We focus on long-time prediction task. The $k$ denotes the number of neighbors used in constructing mesh edges with kNN method. For bilateral stamping with two deformable solids, under the same target, the left column represents the results for the metal, while the right column represents the results for the rubber.

| ABLATIONS | TISSUE MANIPULATION | BILATERAL STAMPING | | | | | |
|---|---|---|---|---|---|---|---|
| | GEOMETRY | GEOMETRY | | STRESS | | PEEQ | |
| W/O EDGES | 0.0269 | 0.0063 | 0.0059 | 0.0292 | 0.1097 | 0.2962 | 0.2429 |
| $k = 2$ | 0.0263 | 0.0058 | 0.0053 | 0.0289 | 0.1063 | 0.2843 | 0.2338 |
| $k = 3$ | 0.0265 | **0.0056** | **0.0051** | 0.0283 | 0.1058 | 0.2870 | 0.2281 |
| $k = 4$ | **0.0253** | 0.0057 | 0.0052 | **0.0278** | **0.1039** | **0.2817** | 0.2265 |
| $k = 5$ | 0.0257 | 0.0057 | 0.0054 | 0.0282 | 0.1049 | 0.2836 | 0.2269 |
| $k = 6$ | 0.0260 | 0.0056 | 0.0051 | 0.0279 | 0.1042 | 0.2842 | **0.2258** |
| $k = 7$ | 0.0254 | 0.0059 | 0.0051 | 0.0284 | 0.1046 | 0.2818 | 0.2294 |
| $k = 8$ | 0.0266 | 0.0062 | 0.0055 | 0.0293 | 0.1072 | 0.2905 | 0.2312 |
| W/O LOADS | 0.0346 | 0.0245 | 0.2353 | 0.0378 | 0.1949 | 0.5441 | 0.5720 |

# G. Out-of-distribution Generalization

In Table 5, we have provided part results of the out-of-distribution experiment on unseen material parameters. We include the complete results in Table 9. It is impressive that Unisoma can still achieve the best performance in the out-of-distribution setting across all target physical quantities. This comes from special explicit modeling design, which enables Unisoma to capture more foundational physical interactions and generalize to unseen material parameters better.

*Table 9.* Complete results of the OOD generalization experiment on the cavity extruding dataset. Relative L2 is recorded. "Outer", "Middle", and "Inner" refer to each deformable solid of the multi-layered cavity from the outside to the inside, respectively.

| | OUTER | | | MIDDLE | | | INNER | |
|---|---|---|---|---|---|---|---|---|
| | GEOMETRY | STRESS | PEEQ | GEOMETRY | STRESS | PEEQ | GEOMETRY | STRESS |
| GEO-FNO | 0.0117 | 0.1036 | 0.1771 | 0.0177 | 0.0790 | 0.1171 | 0.0198 | 0.2526 |
| GINO | 0.0715 | 0.2003 | 0.7461 | 0.1165 | 0.1900 | 0.7268 | 0.1015 | 0.4645 |
| GNO | 0.0238 | 0.1452 | 0.2811 | 0.0444 | 0.1149 | 0.2914 | 0.0449 | 0.3727 |
| LSM | 0.0249 | 0.1194 | 0.3164 | 0.0436 | 0.0840 | 0.2633 | 0.0339 | 0.3111 |
| GALERKIN | 0.0931 | 0.1996 | 0.7899 | 0.1333 | 0.7641 | 0.1398 | 0.1398 | 0.5165 |
| FACTFORMER | 0.0202 | 0.1116 | 0.2289 | 0.0269 | 0.0738 | 0.1355 | 0.0308 | 0.2739 |
| OFORMER | 0.0134 | 0.1064 | 0.1930 | 0.0204 | 0.0722 | 0.1276 | 0.0229 | 0.2596 |
| ONO | 0.0213 | 0.1089 | 0.2088 | 0.0357 | 0.0792 | 0.1642 | 0.0372 | 0.2511 |
| TRANSOLVER | 0.0162 | 0.1066 | 0.2069 | 0.0334 | 0.0765 | 0.1521 | 0.0336 | 0.2643 |
| GRAPHSAGE | 0.0211 | 0.1434 | 0.2594 | 0.0339 | 0.1126 | 0.1971 | 0.0416 | 0.3571 |
| MGN | 0.0178 | 0.1413 | 0.2359 | 0.0280 | 0.1113 | 0.1788 | 0.0385 | 0.3526 |
| **UNISOMA** | **0.0077** | **0.0994** | **0.1527** | **0.0157** | **0.0706** | **0.1125** | **0.0179** | **0.2348** |

*Table 10.* Key hyperparameters and parameter numbers of models.

| | | Deforming Plate | Cavity Grasping | Tissue Manipulation | Bilateral Stamping | Unilateral Stamping |
|---|---|---|---|---|---|---|
| GeoFNO | MODES | [8, 8, 8] | [8, 8, 8] | [8, 8, 8] | [8, 8, 8] | [10, 10, 10] |
| | HIDDENS | 24 | 28 | 28 | 36 | 36 |
| | PARAMETER (M) | 1.30 | 1.77 | 1.77 | 2.92 | 5.47 |
| GINO | MODES | [8, 8, 8] | [8, 8, 8] | [8, 8, 8] | [8, 8, 8] | [8, 8, 8] |
| | HIDDENS | 8 | 8 | 8 | 12 | 18 |
| | LATENT GEOMETRY | 8 | 20 | 20 | 8 | 8 |
| | PARAMETER (M) | 1.41 | 1.41 | 1.41 | 2.70 | 5.61 |
| GNO | HIDDENS | 96 | 96 | 96 | 64 | 64 |
| | KERNEL WIDTH | 128 | 128 | 128 | 128 | 128 |
| | DEPTH | 3 | 3 | 3 | 3 | 3 |
| | PARAMETER (M) | 1.14 | 1.23 | 1.23 | 0.56 | 0.56 |
| LSM | HIDDENS | 8 | 8 | 8 | 16 | 16 |
| | TOKENS | 8 | 8 | 8 | 8 | 8 |
| | BASIS | 12 | 12 | 12 | 12 | 12 |
| | PARAMETER (M) | 1.49 | 1.49 | 1.49 | 5.94 | 5.94 |
| Galerkin | HIDDENS | 128 | 128 | 128 | 192 | 256 |
| | FEEDFORWARD | 512 | 512 | 512 | 512 | 512 |
| | LAYERS | 6 | 8 | 8 | 8 | 10 |
| | PARAMETER (M) | 1.20 | 1.60 | 1.60 | 2.80 | 5.35 |
| Factformer | HIDDENS | 128 | 128 | 128 | 256 | 256 |
| | HEAD DIM | 64 | 64 | 64 | 64 | 128 |
| | KERNEL MULTIPLIER | 3 | 4 | 4 | 4 | 4 |
| | PARAMETER (M) | 1.19 | 1.58 | 1.58 | 3.16 | 6.32 |
| Oformer | HIDDENS | 128 | 128 | 128 | 128 | 128 |
| | DEPTH | 2 | 2 | 2 | 4 | 3 |
| | PARAMETER (M) | 1.48 | 1.48 | 1.48 | 2.63 | 2.06 |
| ONO | HIDDENS | 128 | 128 | 128 | 256 | 256 |
| | LAYERS | 4 | 5 | 5 | 4 | 4 |
| | MLP RATIO | 4 | 4 | 4 | 2 | 4 |
| | PARAMETER (M) | 1.31 | 1.65 | 1.65 | 3.27 | 5.13 |
| Transolver | HIDDENS | 128 | 128 | 128 | 128 | 128 |
| | LAYERS | 12 | 12 | 12 | 12 | 16 |
| | MLP RATIO | 1 | 1 | 1 | 4 | 4 |
| | SLICE | 32 | 32 | 32 | 32 | 32 |
| | PARAMETER (M) | 1.44 | 1.44 | 1.44 | 3.81 | 5.07 |
| MGN | HIDDENS | 128 | 128 | 128 | 128 | 128 |
| | MLP LAYERS | 2 | 2 | 2 | 3 | 4 |
| | PASSING STEPS | 15 | 15 | 15 | 15 | 20 |
| | PARAMETER (M) | 2.32 | 2.32 | 2.32 | 2.88 | 4.51 |
| GraphSAGE | HIDDENS | 256 | 256 | 256 | 256 | 512 |
| | LAYERS | 8 | 8 | 8 | 12 | 10 |
| | PARAMETER (M) | 1.18 | 1.18 | 1.18 | 2.10 | 5.46 |
| Unisoma | PARAMETER (M) | 0.88 | 1.42 | 1.42 | 2.82 | 5.67 |

