# OpenReview forum: "Unisoma: A Unified Transformer-based Solver for Multi-Solid Systems"
_ICML.cc/2025/Conference — ICML 2025 poster_

### Official Review · Reviewer_YnTU · 2025-03-09

**Overall Recommendation:** 3

**Summary:**

This paper proposes an explicit modeling approach for solving multi-solid problems, incorporating factors influencing solid deformation through specialized modules. The authors design a novel transformer-based architecture to achieve this, demonstrating state-of-the-art performance across multiple datasets.

**Claims And Evidence:**

Yes

**Essential References Not Discussed:**

N/A

**Experimental Designs Or Analyses:**

The experimental evaluation could be strengthened by including scaling studies to demonstrate the method's scalability. Additionally, visualizations of the learned Edge Augmented Physics-Aware Tokens would help verify whether the model effectively captures the underlying physical information in the datasets.

**Methods And Evaluation Criteria:**

Yes

**Other Comments Or Suggestions:**

N/A

**Other Strengths And Weaknesses:**

N/A

**Questions For Authors:**

N/A

**Relation To Broader Scientific Literature:**

While this paper focuses specifically on multi-solid scenarios, its methodological contributions may provide valuable insights for the broader field of PDE solving.

**Theoretical Claims:**

This paper does not present any theoretical claims.

---

> ### Author Rebuttal · Authors · 2025-04-01
>
> Sincerely thanks for insightful comments.
>
> > The experimental evaluation could be strengthened by including scaling studies to demonstrate the method's scalability.
>
> Thanks for the insightful suggestion. We conducted a series of scaling experiments to analyze the model’s behavior under varying configurations. Specifically, we evaluated performance across different numbers of slices, processors, channels, and training samples. The default settings used in the paper is: 32 for slice number, 2 for processor number, 128 for channel number and 1000 training samples. We only adjust one variable in one experiment and keep other variables unchanged. The relative L2 recorded in the table is the average accuracy of all physical quantities.
>
> From the results, we observe that increasing the number of slices and channels leads to improved performance up to a certain threshold (approximately slices ≥ 128, channels ≥ 192), beyond which the gains become marginal. This suggests that our model is parameter-efficient, achieving strong performance without requiring excessive capacity. Interestingly, the model shows little sensitivity to depth, with performance remaining largely stable as the number of processors increases. We attribute this to the model's inherently wide architecture, where each layer consists of multiple parallel attention-based modules. This design reduces the model's reliance on depth and makes it less prone to overfitting with increasing layers. Additionally, the relatively flat performance curve across depth may indicate that the model is reaching saturation under the current data regime.
>
>
> |Slice Number|16| 32     | 64     | 96     | 128    | 160    | 192    | 224    | 256    |
> |-|-|-|-|-|-|-|-| ---- | ------ |
> | Relative L2 | 0.1140 | 0.1084 | 0.1078 | 0.1087 | 0.1069 | 0.1063 | 0.1066 | 0.1077 | 0.1073 |
> | Params (M)   | 2.83   | 2.85   | 2.88   | 2.91   | 2.95   | 3.00   | 3.04   | 3.09   | 3.15   |
>
> | Processor Number | 1      | 2      | 3      | 4      | 5      | 6      | 7      | 8      |
> | - | -| - | - | - | - | - | - | - |
> | Relative L2  | 0.1087 | 0.1084 | 0.1087 | 0.1080 | 0.1077 | 0.1069 | 0.1081 | 0.1074 |
> | Params (M)   | 1.82   | 2.85   | 3.87   | 4.89   | 5.92   | 6.94   | 7.96   | 8.98   |
>
> | Channel Number | 64     | 96 | 128 | 160    | 192    | 224    | 256    | 320    | 384    | 448    | 512    |
> |-|-|-|-|-|-|-|-|-|-|-|-|
> | Relative L2| 0.1095 | 0.1084 | 0.1084 | 0.1072 | 0.1070 | 0.1087 | 0.1081 | 0.1124 | 0.1112 | 0.1106 | 0.1109 |
> | Params (M)     | 0.73   | 1.61   | 2.85   | 4.43   | 6.36   | 8.63   | 11.26  | 17.55  | 25.24  | 34.32  | 44.80  |
>
> To further investigate this, we conducted data scaling experiments. The results clearly show that model performance consistently improves as the number of training samples increases. Even at the largest scale we tested (1000 samples), the performance continues to rise, indicating that the model has not yet saturated and remains highly data-efficient.
>
> |Training Samples|200|400|600|800|1000|
> |-|-|-|-|-|-|
> | Relative L2   | 0.1387 | 0.1263 | 0.1210 | 0.1103 | 0.1084 |
>
> Overall, these findings highlight the robustness and scalability of our model. It delivers competitive accuracy with moderate parameter counts, maintains stability across architectural depths, and continues to benefit from additional data—demonstrating strong practical potential in real-world applications.
>
> > Visualizations of the learned Edge Augmented Physics-Aware Tokens would help verify whether the model effectively captures the underlying physical information in the datasets.
>
> To evaluate the effectiveness of the **Edge Augmented Physics-Aware Tokens**, we visualize the slice weights of two deformable solids in the **Bilateral Stamping** scenario. The comparison is available at https://anonymous.4open.science/r/unisoma_icml_2025-3DEF
>
> As shown in the figures, a **horizontal comparison** reveals that **different slices attend to different spatial regions after projecting from the original mesh space**: some focus on the pressed area under the rigid solid, others on the central stretching region, and some on the fixed ends. This indicates that the tokens successfully group points with similar physical states and differentiate between slices, enabling the model to capture diverse underlying physical behaviors.
>
> More importantly, in the **vertical comparison** between orderly corresponding slices of the two deformable solids, we observe that **they tend to focus on similar regions**—areas that are not only of high relevance to each solid individually but are also **likely to come into contact**. This alignment further supports the effectiveness of Edge Augmented Physics-Aware Tokens in capturing meaningful, structured physical interactions.
>
> Thank you for your positive recognition of our work. We have carefully addressed the concerns you raised and provided additional explanations and experimental results to support our claims. We look forward to your response and feedback.

---

### Official Review · Reviewer_skRm · 2025-03-17

**Overall Recommendation:** 3

**Summary:**

The paper explicitly models the contact constraints and loads in multi-solid systems, using a Transformer-based framework.
- The system contains three types of objects, deformable solids, rigid solids, and forces. Instead of treating each point as a token, the paper proposes to incorporate the mesh edges into embedding and proposes edge-augmented tokens.
- Then, stacked processors are used to model physical interactions among solids. The processor separately consider and contact constraints, the loads, and the effects on each deformable solid. An independent deformation module is used for each deformable solid.
- The outputs of the processor are mapped back to the original domain by weighted broadcast, to get the predicted outcomes.

The proposed methods is evaluated on two tasks, long-time prediction and autoregressive simulation.
- The experiments use 7 datasets, including different systems containing few solids or multiple solids in 3D spaces. 4 of the datasets are public datasets for autoregressive task. The authors construct 3 additional datasets.
- The proposed method is compared with ten baselines, and achieves state-of-the-art performance across the benchmarks.



update after rebuttal
Thank the authors for the responses. I decide to keep my original score

**Claims And Evidence:**

yes.

**Essential References Not Discussed:**

No.

**Experimental Designs Or Analyses:**

yes.

**Methods And Evaluation Criteria:**

yes.

**Other Comments Or Suggestions:**

I just have some additional questions listed below.

**Other Strengths And Weaknesses:**

The explicit modeling of contact constraints, loads, and deformable modules using transformer-based models seems novel to me. Additionally, the paper presents good experimental results, comparing against multiple baselines across a relatively large set of tasks.

**Questions For Authors:**

- How do you get all the solids pairs that are likely to contact when calculating contact constraint?
- As mentioned in the paper, the system's input includes deformable solids, rigid solids, and loads. Given a specific system, how do you get the "loads" input objects?
- In the autoregressive task, the framework directly learns to predict the next step. Is it easy or hard to generalize to different time steps?
- The attention module has a quadratic time complexity regarding the number of tokens. As the number of input tokens increases, computation can become costly. Does the proposed method (which applies attention on the sliced inputs) face a similar issue? Could you provide a brief complexity analysis?

**Relation To Broader Scientific Literature:**

Compared to the existing works, this paper explicitly models the multi-solid systems using transformer-based frameworks instead of implicit models.

**Theoretical Claims:**

N/A

---

> ### Author Rebuttal · Authors · 2025-04-01
>
> Sincerely thanks for insightful comments.
>
> > How to get all the solids pairs that are likely to contact?
>
> In most cases, such as the stamping and grasping scenarios discussed in the paper, the solid pairs that are likely to come into contact are **known a priori** based on the deterministic nature of the physical setup.
>
> For systems where contact is less certain, as mentioned in Line 368-375, we include **solid pairs with a high likelihood of contact** in the contact module. The **adaptive interaction allocation** mechanism then controls the extent to which each contact influences the deformation of a solid, effectively weighting more relevant interactions. This makes the process **flexible and robust** across both well-defined and uncertain contact settings.
>
> > Given a specific system, how to get the "loads"
>
> As a supplement to lines 127–130 in the paper, we clarify that for a moving solid, We treat the displacement $d = u(t+1) - u(t)$ as a **load** applied over that time interval—commonly referred to as a **displacement load** [1]. Its positions at time $t$ and $t+1$ are denoted as $u(t)\in R^{N\times 3}$ and $u(t+1)\in R^{N\times 3}$, respectively. We represent each load in a **Lagrangian description** as the concatenation $\text{concat}(u(t), d)\in R^{N\times 6}$, encoding both its origin position and next movement. When the moving solid comes into contact with others, this displacement load is transferred as a **force** onto the contacting objects, influencing their deformation.
>
> > Is it easy or hard to generalize to different time steps in the autoregressive task?
>
> Similar to MGN[2], our framework learns to predict **the next state given the current state** in an autoregressive manner. This step-wise prediction scheme does not explicitly encode time intervals, so generalization across different time steps depends on the **temporal distribution of the training data** and the model’s capacity to learn the underlying dynamics.
>
> To evaluate the generalizability across time steps, we designed an experiment using the **Cavity Grasping** dataset:
> 1. We directly test Unisoma, trained on original time step size (600 samples, 100 epochs), on data with **doubled time intervals**.
> 2. We **fine-tune** (20 epochs) the same model using a small amount of doubled-step data (120 samples), then test on data with doubled time intervals.
> 3. We perform **full training** (600 samples, 100 epochs) on the doubled-interval data and test accordingly.
>
> In all cases, all other parameters and test data remain consistent)
>
> ||1.Directly test|2.Fine-tuning|3.Full training|
> |-|-|-|-|
> |Rollout-all RMSE($10^{-3}$)|13.43|11.06|9.68|
>
> These results reveal several insights:
>
> - The **drop in performance (higher RMSE)** when directly applying the model to doubled-step data indicates **limited generalization** when time step statistics shift significantly.
> - However, **a small amount of fine-tuning** brings considerable improvement, suggesting that the model retains useful representations of the dynamics that are **transferable** across step sizes.
> - **Full retraining** yields the best performance, as expected, since the model can directly adapt its dynamics modeling to the new time scale.
>
> > About the complexity of Unisoma
>
> While attention mechanisms indeed have **quadratic time complexity** with respect to the number of tokens, our attention operations in Eq. (2) and Eq. (6) are performed over the slices, not directly on the original mesh points. Each slice is corresponding to an **Edge Augmented Physics-Aware Tokens**. Usually, the number of slices is significantly smaller than the number mesh points.
>
> Let the number of input mesh points be $N$, which are converted to $M$ slice tokens via Eq. (1) and $M \ll N$. This step has a complexity of $\mathcal{O}(MNC)$. The attention in Eq. (2) and Eq. (6) is then applied to sequences of length $M$, with a complexity of $\mathcal{O}(M^2C)$ and is irrelevant to the mesh points $N$. Moreover, the number of modules per layer (contact modules, deformation module, etc) is typically small and the cost is negligible compared to the embedding step. Therefore, the **overall complexity is approximately $\mathcal{O}(MNC + M^2C)$**. This means that as the input sequence length $N$ increases, the computational cost of our model scales **almost linearly** ($M \ll N$ and $M$ is usually fixed), making it significantly more efficient than standard attention over full mesh sequences.
>
> Thank you for your positive recognition of our work. We have carefully addressed the concerns you raised and provided additional explanations and experimental results to support our claims. We look forward to your response and feedback.
>
> [1] Mau S T. Introduction to structural analysis: displacement and force methods[M]. Crc Press, 2012.
>
> [2] Pfaff T, Fortunato M, Sanchez-Gonzalez A, et al. Learning mesh-based simulation with graph networks[C]//International conference on learning representations. 2020.

---

### Official Review · Reviewer_M6EV · 2025-03-17

**Overall Recommendation:** 2

**Summary:**

This paper presents a transformer-based framework for explicitly modeling multi-solid interactions. The approach differs from implicit approaches that merge solids into a unified PDE or use graph-based message passing, and presents an explicit modeling one that structures interactions through a deformation triplet of a deformable solid, an equivalent load, and an equivalent contact constraint. It employs contact modules, an adaptive interaction allocation mechanism, and a deformable module to capture and process deformations. The model is evaluated on seven datasets and two multi-solid tasks, demonstrating improvements over existing deep learning methods in long-time prediction and autoregressive simulation.

**Claims And Evidence:**

Overall, the paper presents strong empirical evidence supporting its claims, particularly regarding Unisoma's accuracy, efficiency, and ability to handle multi-solid interactions.

**Essential References Not Discussed:**

The paper claims to be the first at explicit modeling for multi-solid systems, but similar ideas have been explored before. "NCLaw: Learning Neural Constitutive Laws From Motion Observations for Generalizable PDE Dynamics" (ICML 2023) also integrates explicit modeling by enforcing known PDE structures while learning constitutive models, ensuring physical correctness and generalizability.

**Experimental Designs Or Analyses:**

I think the experimental designs and analyses are sound. The paper compares Unisoma against strong baselines across multiple datasets using relevant tasks and metrics. It also includes out-of-distribution testing and efficiency comparisons.

**Methods And Evaluation Criteria:**

I believe the methods and evaluation criteria are genreally well-designed for the problem. The chosen benchmarks, tasks, and metrics effectively demonstrate Unisoma’s strengths in multi-solid simulation across different scenerios.

**Other Comments Or Suggestions:**

See above comments

**Other Strengths And Weaknesses:**

The paper is weak in explaining its key method and is difficult to read in its current form. Several sections lack clarity, making it challenging to understand the approach. The explanation of the slicing algorithm and its connection to contact modeling is particularly vague, and the use of symbols and hyperparameters without proper definitions adds to the confusion. The training procedure and loss functions are not well-documented in the main text, making it unclear how supervision ensures physically meaningful constraints. Additionally, while the paper claims to use explicit modeling, it does not clarify how each module enforces physical correctness, particularly for contact constraints. While the experimental results apepar to be strong, these issues obscure the main contributions of the paper and make it unnecessarily complex.

**Questions For Authors:**

1. Line 178-185: The explanation of k-NN, edge sets, and deep features is unclear. What exactly are the x values referred to as "deep features"? What does C represent? How is the edge set E computed, and what is k in this context? How is it chosen?

2. Slicing Algorithm: Figure 3 does not clearly explain how slicing is performed. How does the slicing process apply to a general mesh? Why is the slice domain relevant to contact modeling? How does slice composition help distinguish contact interactions from other types of interactions?

3. Equations 2 and 6: How are Q, K, and V computed in these equations? What is their role in the model, and how does the attention mechanism specifically capture physical interactions?

4. Training and Loss Functions: The loss functions are vaguely described in the appendix. Can the authors provide a clearer breakdown of the loss functions used for each task? How does the loss ensure that each explicit module (deformation, load and contact) learns its intended physical meaning?

5. Physical Meaning of the Contact Module: How do the authors verify that the contact module correctly handles contacts? Does the contact module account for nonlinear contact forces, friction, or material-dependent constraints, or does it treat all contacts as uniform interactions?

**Relation To Broader Scientific Literature:**

The paper builds on implicit modeling methods like PINNs and Neural Operators, as well as graph-based approaches using message passing such as MGN. Unlike these, this paper adopts explicit modeling, aligning with traditional FEM while leveraging Transformer-based PDE solvers for efficiency.

**Theoretical Claims:**

The paper does not contain formal mathematical proofs for theoretical claims.

---

> ### Author Rebuttal · Authors · 2025-04-01
>
> Sincerely thanks for insightful comments.
>
> > Weakness: enforcing physical correctness
>
> We first emphasize that the “explicit modeling” we defined **lies not in the explicit PDE constraints used in PINNs or hybrid models**, but rather in **how the model structure leverages and organizes input information**. In PINNs and hybrid models (e.g., NCLaw), PDEs are added to the loss to guide training, often relying on strong assumptions such that explicit PDEs exist—e.g., NCLaw assumes elastodynamic behavior. In **multi-solid systems**, it is difficult to define a single global and well-studied PDE (like N-S equation in CFD) to describe the entire system accurately. Instead, the behavior is usually governed by multiple **local relations**, such as contact penalties, load applications and deformations. Directly embedding them into the loss leads to **complex, multi-term objectives** that are hard to optimize and often limited in applicability. This remains an **open challenge** in the field.
>
> Therefore, we take a **purely data-driven approach** and propose to model the physics **through the model architecture**. Our framework explicitly organizes data using structured modules to capture the underlying physics. This design allows effective learning across **diverse materials, object counts, and task types**, with consistently strong results. We hope this view can complement PDE-based approaches, and we look forward to future work that combines both in a unified framework.
>
> > similar explicit modeling ideas
>
> While NCLaw is a valuable hybrid model combining neural networks with traditional solvers, our method is **purely data-driven**, learning physical priors **through architectural biases**. They belong to different classes, as NCLaw depends on **classical PDE solvers**.
>
> >  Line 178-185
>
> The x denotes the deep features of an input object. For example, given a rigid solid $u^r_i \in \mathbb{R}^{N_i^r \times C_r}$, we project it using a linear layer: x = Linear($u^r_i)$, where $x \in \mathbb{R}^{N_i^r \times C}$. This is applied to each object individually, with a unified feature dimension $C$. We will clarify this more clearly in the revision. The edge set $E$ is constructed by k-nearest neighbors on mesh points of each object individually, i.e., $E = \text{kNN}(u^r_i)$, where each point connects to its $k$ nearest neighbors. As shown in Appendix F, we find $k = 3 \sim 5$ yields better performance.
>
> > Slicing
>
> As shown in Eq. (1), slice weights are computed on deep features $x$ via a linear layer and softmax, and edge weights are derived from connected point weights. The slice is then formed by aggregating point and edge features. Since $x$ comes from mesh points, this applies directly to general meshes.
>
> As noted in Line 201, slices group points with similar physical states, enabling attention to capture **physically consistent interactions** (Remark 3.1 in Transolver). We will emphasize this more clearly in the revision. When composing a slice from two potentially contacting solids, **contact becomes important interactions**, while unrelated interactions are suppressed. This focus improves attention modeling and enhances contact capture. Our results confirm that this explicit structure significantly boosts accuracy.
>
> > Equations 2, 6
>
> Q, K, and V are computed using standard Transformer linear projections: $Q = W_q(g), K = W_k(g), V = W_v(g)$, with learnable matrices $W_q, W_k, W_v$. In Eq. (2) and (6), $x$ is the **composed slice** from contacting solids or deformation triplets. As mentioned in the previous response, attention in the slice domain enables the model to learn **physically consistent interactions**, containing contact and deformation.
>
> > Training and Loss
>
> We apologize for the brevity due to space constraints. As shown in Appendix C, our loss functions follow standard setups in related works (e.g., relative L2 for long-time prediction, RMSE for autoregressive simulation). As noted in the first response and Lines 076–089, Unisoma does not use PDE-based losses but **learns physical interactions structurally**:  the contact module, adaptive interaction allocation, and deformation triplet, all built on slice composition. These modules are trained end-to-end, and their physical roles are learned through architecture design.
>
> > Physical Meaning of the Contact Module
>
> Unlike PINNs or hybrid models that use PDEs as explicit constraints, we treat **contact interactions as learnable physical relationships** encoded in **high-dimensional features**. The model learns to minimize loss by focusing on key interactions, especially contact. We verify its effectiveness through improved accuracy across multiple datasets, particularly in complex multi-contact scenarios, where baselines that model interactions holistically tend to suffer performance degradation due to the holistic interaction mixture. Our structured, explicit modeling helps avoid such issues by organizing information around physically meaningful groupings.

---

### Official Review · Reviewer_g8HG · 2025-03-17

**Overall Recommendation:** 3

**Summary:**

This paper focuses on multi-solid tasks and proposes Transformer-based model to deal with the interactions between objects.
To better handle the interactions, the paper proposes to explicitly model the external forces and contact interactions, whose hidden representations are combined with the objects’ embeddings to predict the physical quantities. Experiments show that the proposed method outperforms baselines on various domains.

## Update after rebuttal
I appreciate the authors’ efforts in addressing my concerns, and the lastest explanation for "Theoretical Claims" is reasonable to me. Therefore, I have updated my score to 3.

Additionally, if the paper is accepted, it would be beneficial to include experiments demonstrating the performance of the corrected formulation with learnable parameters, accompanied by any necessary discussion.

**Claims And Evidence:**

The claims are generally clear.

**Essential References Not Discussed:**

The author should discuss TIE [1], which is also a Transformer-based model focusing on simulation and includes performance on multi-solid systems.

[1]. Shao, et al. Transformer with Implicit Edges for Particle-based Physics Simulation, ECCV2022.

**Experimental Designs Or Analyses:**

1. I notice that in most of the quantitative results, such as Table 1, there are more baselines. However, for autoregressive simulation task (Table 3) and the efficiency comparisons (Table 5), fewer baselines are compared and the baselines are not the same as those in Table 1. For example, new baselines like HOOD and HCMT appear only in Table 3 while missing in Table 1. Could the author provide more details about how to choose these baselines and the complete tables of quantitative comparisons for all baselines?
2. In the appendix at L787-803, the author claims that they “use 1200 samples” in the RiceGrip domain. However, in the original repo of DPI-Net [1], only 5 samples are publicly available. How the 1200 samples are obtained?
3. The task of “long-time prediction” seems less convincing. From my understanding, the author may try to demonstrate that the model has robust performance in long-term predictions, which is also mentioned in MGN [2]. However, the settings  of “long-time prediction” as mentioned in Section 4.2 seem to predict the last frame given the initial frames, which are different from the experiments in MGN. On the other hand, the settings of “autoregressive simulation task” are much closer to the settings in MGN, where the total number of frames should be large enough. Since long-term prediction is an important challenge in simulation, more results should be provided. For example, results of longer trajectories should be provided. Notice that RiceGrip only has 41 frames for each sequence, it is not considered long enough from my opinion.
4. While the video results may be optional, they are extremely important to evaluate the dynamic results of simulation, since this is a task predicting dynamics instead of static scenes. The problems, such as long-term predictions, can be easily observed in the video results. The same applies to the improvement.  However, this paper does not provide video results, making the experiments less convincing.

[1]. Li, et al. Learning particle dynamics for manipulating rigid bodies, deformable objects, and fluids. ICLR2019.
[2]. Pfaff, et al. Learning mesh-based simulation with graph networks. ICLR2021.

**Methods And Evaluation Criteria:**

The methods and criteria make sense.

**Other Comments Or Suggestions:**

Please refer to "Questions For Authors".

**Other Strengths And Weaknesses:**

Please refer to "Questions For Authors".

**Questions For Authors:**

Here I summarize all the concerns:
1. Questions about the Equation 4 and 9. (Theoretical Claims)
2. Incomplete baselines comparisons. (Experimental Designs Or Analyses Q1)
3. Questions about RiceGrip data. (Experimental Designs Or Analyses Q2)
4. Autoregressive simulation task with longer trajectories as “Long-time predictions” could be more convincing. (Experimental Designs Or Analyses Q3)
5. Missing video results. (Experimental Designs Or Analyses Q4)
6. Missing reference. (Essential References Not Discussed)

While I may be positive to this work, I hope the author could carefully address my concerns.

**Relation To Broader Scientific Literature:**

N/A.

**Theoretical Claims:**

Equation 4 and 9 seem questionable. Since the author claims that $\sum w_{i,j}^{\alpha} \approx \sum w_{i,j}^{\beta} \approx 1$, a simple case to verify Equation 4 is that: if we just choose $\sum w_{i,j}^{\alpha} = \sum w_{i,j}^{\beta} = 1$, then the first row in Equation 4 becomes $\frac{\sum w^{\alpha} \mathbf{x}^{\alpha}}{\sum w_{i,j}^{\alpha}} + \frac{\sum w^{\beta} \mathbf{x}^{\beta}}{\sum w_{i,j}^{\beta}} = \sum w^{\alpha} \mathbf{x}^{\alpha}+\sum w^{\beta} \mathbf{x}^{\beta}$, while the second row of Equation 4 becomes $\frac{\sum w^{\alpha} \mathbf{x}^{\alpha}+\sum w^{\beta} \mathbf{x}^{\beta}}{\sum w_{i,j}^{\alpha}+\sum w_{i,j}^{\beta}}  = 0.5(\sum w^{\alpha} \mathbf{x}^{\alpha}+\sum w^{\beta} \mathbf{x}^{\beta})$. Obviously, these two equations vary a lot and cannot be connected by $\approx$ in Equation 4. The same applies to Equation 9. Could the author provide further explanations or corrections?

---

> ### Author Rebuttal · Authors · 2025-04-01
>
> Sincerely thanks for insightful comments.
>
> > Equations 4, 9
>
> Thanks for pointing this out—Equations (4) and (9) do contain errors, and we apologize for the typos. We give a mathematically correct formulation here:
> $$
> z^\xi_j=\frac{\sum_{i=1}^{N^\alpha} w^\alpha_{i,j}x^\alpha_i + \sum_{i=1}^{N^\beta} w^\beta_{i,j}x^\beta_i}{\sum_{i=1}^{N^\alpha}w_{i,j}^\alpha +\sum_{i=1}^{N^\beta}w_{i,j}^\beta}=\frac{(\sum_{i=1}^{N^\alpha}w^\alpha_{i,j})z_j^{\alpha} + \sum_{i=1}^{N^\beta} (w^\beta_{i,j})z_j^{\beta}}{\sum_{i=1}^{N^\alpha}w_{i,j}^\alpha + \sum_{i=1}^{N^\beta}w_{i,j}^\beta}=\theta z_j^{\alpha}+(1-\theta)z_j^{\beta}
> $$
> We make $\theta$ learnable. We test the correct form but it yielded **no noticeable performance gain** over direct element-wise addition. Hence, we chose **direct element-wise addition** for slice composition due to its simplicity, efficiency, and performance.
>
> The **slice composition** is a simple yet effective fusion strategy that works well in practice. Importantly, it allows each solid to maintain its **own slice projection parameters**, rather than concatenating the two solids and projecting them jointly within each contact module (without parameter sharing across modules). We experiment with this more complex approach, but it results in **no significant improvement**, despite parameters increasing.
>
> We will correct Equations (4) and (9) and clarify this design in the revision. Once again, we sincerely apologize for it and thanks for the careful observation.
>
> > Selection of baselines
>
> We first emphasize the tasks. As claimed in the **bold text and corresponding citations in Section 3**, our tasks contain:
>
> - **Long-time prediction** follows Transolver, where the model directly predicts the target state from the given state and loads, usually spanning long time and skipping intermediate steps (e.g., using the 1st frame to predict the 105th frame in Cavity Grasping).  It can be formulated as: $P(\hat{x}(t+T)|x{(t)})$, where $T$ spans many time steps. This task needs **long-range dependence** and is important for fast inference in many industrial scenarios..
>
> - **Autoregressive simulation** aligns with MGN, focusing on step-by-step rollout.  It can be formulated as: $P(\hat{x}(t+1)|x(t))$, $P(\hat{x}(t+2)|\hat{x}(t+1))$, …, $P(\hat{x}(t+T)|\hat{x}(t+T-1))$. It focus more on the rollout trajectory and is suitable for scenarios where intermediate states are essential.
>
> We select baselines based on their **recency and relevance to the tasks**. For long-time prediction, we compare Unisoma with **prevalent domain-wise and GNN-based models**. Domain-wise methods are better suited for global inference, while GNNs like MGN suffer from limited receptive fields, though we still include them for completeness. For autoregressive simulation, the task is mainly addressed by GNNs (e.g., MGN, HCMT, HOOD), and domain-wise models have rarely been extended to it. Thus, we focus on the most competitive baselines. For the efficiency comparison (Table 5), we selected models that are highly efficient or widely used in operator learning. GINO and GNO represent domain-wise neural operator models, while OFormer, ONO, and Transolver are recent efficient linear attention-based solvers.
>
> We provide full efficiency comparison here. Notably, due to the difference in input shapes between Euler [B, X, Y, Z, C] and Lagrange [B, (X*Y\*Z), C] views, the former is generally more memory-friendly for the same number of points (regular grid), but less suitable for solid system. We use a view mapping for part of baselines (Line 992-999). The main paper compares Lagrange-compatible models and we include Euler-only models (tagged “E”) here. Despite this disadvantage, **Unisoma maintains high memory efficiency as point count increases**.
>
> |||Bilateral||| Unilateral||
> |-|-|-|-|-|-|-|
> ||Param|Time|Mem|Param|Time|Mem|
> |GeoFNO(E)|2.92| 32.28|0.75|5.47|58.73|1.38|
> |LSM(E)|5.94|43.23|2.77|5.94|225.85|19.49|
> |Galerkin|2.80|104.24|3.89|5.35|570.38|20.65|
> |Factformer(E)|3.16|33.15|1.61|6.32|126.96|14.21|
> |GraphSAGE|2.10|46.36|1.83|5.46|304.89|8.92|
> |MGN|2.88|119.68|13.93|4.51|373.49|23.30|
> |Unisoma|2.85|70.96|0.93|5.21|152.55|1.03|
>
> > RiceGrip Dataset
>
> The DPI-Net repo includes the data generation code scripts. We generate the data on Ubuntu 18.04 with GTX 1080 (CUDA 9.1).
>
> > Long-time prediction and autoregressive simulation
>
> As clarified before in *Selection of baselines*, long-time prediction aims to directly infer the target without intermediate steps, while autoregressive simulation rolls out step-by-step (105 steps for Cavity Grasping/Tissue Manipulation, 120 for Cavity Extruding). These simulation steps align with prior works (e.g., HOOD, HCMT), but **our cases involve more solids** (Cavity Extruding), making them more challenging and less explored.
>
> > Video results
>
> We provide videos at https://anonymous.4open.science/r/unisoma_icml_2025-3DEF
>
> > Discussion of TIE
>
> We acknowledge the relevance of TIE and will include a discussion.

---

> > ### Comment · Reviewer_g8HG · 2025-04-03
> >
> > Thank you for your response and most of my concerns have been addressed. However, I am still worried about the potential impact of the incorrectness as mentioned in “Theoretical Claims”.
> >
> >  I understand that the authors provide results that the correct formulation may not substantially affect the performance. However, if the original claims are incorrect, it would greatly hurt Remark 3.2 at L239-240, and explanations (L240-250) and results related to “slice decomposition” would be less convincing, which may necessitate significant revisions of this paper.
> >
> > I still expect that the author can explain more about this formulation and  the potential impact. Currently, I will maintain my original score.

---

> > > ### Author Response · Authors · 2025-04-03
> > >
> > > We sincerely thank the reviewer for the thoughtful follow-up and the opportunity to clarify the theoretical impact of Eq. (4) and (9) on our framework, particularly regarding **Remark 3.2** and the **definition of slice composition and decomposition**.
> > >
> > > We fully agree that the originally stated Eq.(4) and (9) contained an assumption that could undermine the theoretical soundness of *Remark 3.2*. We have already corrected the formulation in last response, and clarify that slice composition is a mathematically grounded **linear combination** in the shared slice space, where information from multiple solids is projected and aligned on the same slice index. Crucially, this operation does not require **feature concatenation or projection in the original object space**. Instead, it supports structured, interpretable fusion based on slice-wise views.
> > >
> > > More importantly, although we have modified the original equations, **Remark 3.2 remains valid and meaningful.**
> > >
> > > Firstly, **the new definition of slice composition has no influence on slice decomposition**. Slice decomposition allows us to embed each object into slice domain individually, rather than embedding the entire domain as originally defined in Transolver. As shown in Line 252-272, if we embed the whole domain which contains two objects $x_\alpha$ and $x_\beta$, the slice weight is $w=[w_{1,j}^\alpha, w_{2,j}^\alpha,\cdots,w_{N^\alpha,j}^\alpha,w_{1,j}^\beta, w_{2,j}^\beta,\cdots,w_{N^\beta,j}^\beta]$.  Letting $x=\text{concat}(x_\alpha,x_\beta)$, the slicing is:
> > > $$
> > > z_j=\frac{\sum_i^{N^\alpha+N^\beta}w_{i,j}x_i}{\sum_i^{N^\alpha+N^\beta}w_{i,j}}=\frac{\sum_i^{N^\alpha}w^\alpha_{i,j}x^\alpha_i+\sum_i^{N^\beta}w^\beta_{i,j}x^\beta_i}{\sum_{i=1}^{N^\alpha}w_{i,j}^\alpha +\sum_{i=1}^{N^\beta}w_{i,j}^\beta}
> > > $$
> > > Following this formulation, embedding each object individually — e.g., $z_j^\alpha=\frac{\sum_{i=1}^{N^\alpha}w_{i,j}^\alpha x_i^\alpha}{\sum_{i=1}^{N^\alpha}w_{i,j}^\alpha}$ as defined in Eq.(3)—can be viewed as a special case where the slice weights for other objects (e.g., $w^\beta=0$) are 0. In this process, the corrected definition of slice composition does not affect slice decomposition.
> > >
> > > Secondly, the claim in Remark 3.2— the slice composition enables object-aware interaction by preserving each solid’s slice projection independently—is **strengthened by the fact that the composition is a structured linear operation within the slice domain**. We include the **revised version of Remark 3.2** (from left column of Line 272) here and we will revise Appendix.B accordingly.
> > >
> > > “
> > >
> > > Furthermore, we define a new slice domain representation $z^\xi\in\mathbb{R}^{M\times C}$  formulated as:
> > > $$
> > > z^\xi_j=\frac{\sum_{i=1}^{N^\alpha} w^\alpha_{i,j}x^\alpha_i + \sum_{i=1}^{N^\beta} w^\beta_{i,j}x^\beta_i}{\sum_{i=1}^{N^\alpha}w_{i,j}^\alpha +\sum_{i=1}^{N^\beta}w_{i,j}^\beta}=\frac{(\sum_{i=1}^{N^\alpha}w^\alpha_{i,j})z_j^{\alpha} + \sum_{i=1}^{N^\beta} (w^\beta_{i,j})z_j^{\beta}}{\sum_{i=1}^{N^\alpha}w_{i,j}^\alpha + \sum_{i=1}^{N^\beta}w_{i,j}^\beta}\approx\theta z_j^{\alpha}+(1-\theta)z_j^{\beta}
> > > $$
> > > where $\theta$ are learnable parameters. Here, $z^\xi$ is the linear composition of $z^\alpha$ and $z^\beta$, referred to as *slice composition*. Accordingly, the operation in Eq.(3) is termed *slice decompostion*. We first construct multiple pure slice domains during embedding. Through slice composition, we merge two slice domains that are contact-related. In practice, we adopt **direct element-wise addition** in Eq.(2) as a simple, parameter-free realization of this linear combination. This design achieves comparable performance to the learnable form, while reducing complexity. Although this simplified form does not perform explicit averaging (e.g., $0.5(z^\alpha+z^\beta)$), the resulting features are subsequently processed by normalization and attention layers (e.g., in the contact module), which mitigates effects from scale differences. We then apply attention mechanism to capture the physical interaction within the composed slice domain. This avoids information loss and minimizes interference from unrelated objects.
> > >
> > > ”
> > >
> > > In summary, correlated slice composition has **no impact on slice decomposition**, and Remark 3.2 remains theoretically valid under the revised formulation **with minimal required changes**.
> > >
> > > Finally, we thank the reviewer once again for the careful analysis and constructive feedback, which helped us significantly improve the theoretical clarity of our work.

---

### Official Review · Reviewer_ksEb · 2025-03-17

**Overall Recommendation:** 3

**Summary:**

This paper presents the Unisoma focusing on the PDE solving of multi-solid systems. Different from previous methods, Unisoma proposes to embed the solid type (rigid or deformable) and load information into the model for explicit modeling of multi-solid interactions. Technically, Unisoma employs the slice operation proposed by Transolver to learn the physical states of solid information and external load. Additionally, the correlation between different solid units is calculated by the attention mechanism. Finally, attention is applied to compositional features with explicitly embedded solid information. Unisoma performs well in extensive benchmarks with good efficiency and generalizability.

**Claims And Evidence:**

About the statement of Transolver "However, they only consider the point-level features and spatially aggregate points on the whole domain. This leads to the loss of local relationships of mesh points which are important to model local interactions.", I think it is incorrect as in Transolver's official paper, the slice operation can also be conducted based on convolution, which can consider the local information.

**Essential References Not Discussed:**

I think this paper gives a comprehensive discussion of related works.

**Experimental Designs Or Analyses:**

(1)	About the efficiency comparison with Transolver, I think it is unfair as the Transolver has much more layers and MLP ratio than Unisoma, as listed in Table 9. Actually, since Unisoma adopts the same slicing operation proposed by Transolver and additionally captures much more relations and solid information, it is theoretically impossible that Unisoma is more efficient than Transolver. Besides, I think the kNN clustering operations in Line 178 can be very time-consuming. I do not think the current efficiency statistics consider the clustering step.

(2)	About the fairness of comparison, I am wondering if Transolver or other baselines receive the same additional information as Unisoma, e.g. the solid types or loads. Actually, in the formalization of Transolver’s paper, it can also receive the optional physics information as the input. I think the authors should give different methods the same context of input.

**Methods And Evaluation Criteria:**

Yes

**Other Comments Or Suggestions:**

Please see above.

**Other Strengths And Weaknesses:**

## Strengths

1.	This paper presents a reasonable method tailored to the multi-solid systems.
2.	The authors provide comprehensive experiments to verify the model’s efficiency.
3.	This paper is well-written and gives a complete review of previous works.

## Weaknesses
1.	About the categorization of “implicit” and “explicit” modeling.

Actually, since the additional solid information is point-wise, I think all the baselines can receive the same inputs as Unisoma. For example, the authors can just concatenate all the physics information along the channel-wise and project them into deep features. In this way, Transolver or other baselines can also make an explicit modeling of multi-solid systems, since the feature has been attached with “explicit” information.

Thus, I think the current discussion about previous methods is kind of inappropriate.


2.	About the efficiency comparison with Transolver.
As I stated before, I think the current efficiency results are unfair. Please give a detailed explanation. Also, I would like to see the efficiency comparison on other datasets.

3.	Whether different methods use the same physics information as inputs or not? please give a detailed clarification. If not, please input the same physics information to other baselines to enable a fair comparison.

4.	Writing issues.
-	Rethinking the categorization of “implicit” and “explicit” modeling.
-	Please give a citation to Transolver at the beginning of Section 3.2. This suggestion is not only for ensuring a clear discussion of previous work but also for making the slice operation easier to understand.
-	I think the authors should give the definition of some physics concepts, e.g. equivalent load or contact constraints at the beginning of related work, which will make this paper easier to understand.
-	I would suggest the authors use the subscript type of $ N_d, N_f, N_r$, as the superscript is easy to mix with exponential numbers.

I think this paper provides a practical method for multi-solid modeling. However, as I have some concerns about the correctness of experiments, I cannot give a positive score. I would like to see the authors’ response.

**Questions For Authors:**

N/A

**Relation To Broader Scientific Literature:**

This paper is based on the design of Transolver but gives an insightful design tailored to the multi-solid system.

**Theoretical Claims:**

N/A

---

> ### Author Rebuttal · Authors · 2025-04-01
>
> Sincerely thanks for insightful comments.
>
> > About the convolution and local information in Transolver
>
> From the paper and official code of Transolver, the conv is applied only to **structured meshes or uniform grids** (Section 3.1 in Transolver); for **irregular meshes**—which are the focus of us—Transolver uses linear layers. Moreover, in Transolver, when conv is used, it is for generating deep features and slice weights at point level. The projection into slices is still by globally spatial aggregation over all points, diluting local relations. In contrast, as shown in Table 8, our **Edge Augmented Physics-Aware Tokens incorporate neighbor information via mesh edges into the slice projection**, preserving local relations. We will revise the sentence to:
> “However, when tackling irregular meshes, they only consider the point-level features and spatially aggregate points on the whole domain when transforming the mesh points into the slice domain....”
>
> > Efficiency comparison with Transolver
>
> As explained in Appendix E, we ensured fair model capacity by **aligning the total parameter counts** of all baselines (except OOM error), with Unisoma. Efficiency was compared under similar capacity, making the evaluation meaningful. While both Unisoma and Transolver use slice, they differ in real computation. Transolver slices and deslices at **every layer** and applies FFN() **after deslicing on full mesh points**, leading to high memory usage as point count grows.  Unisoma slices/deslices only once and performs **most FFN() within the slice space**, greatly reducing memory.
>
> As explained in Line 406-412, Transolver treats all points as a single large tensor of size $N \times C$, applying slicing and FFN() directly, which leads to high memory usage. In contrast, Unisoma processes each object separately with **smaller tensors $N_i \times C$, where $\sum N_i = N$**. This splits large matrix operations into smaller ones, reducing memory overhead. Frameworks like PyTorch allocate memory for intermediate activations, so large tensors increase peak usage.
>
> We include kNN time at efficiency test, using a KDTree algorithm with O(N log N) complexity. Since kNN is computed per object individually, not over all points, its cost remains small compared to the model’s runtime which is related to the number of all points.
>
> >  Fairness of comparison
>
> As described in Appendix E, we made our best effort to ensure fairness .We aligned the parameter count and carefully tuned each baseline to achieve better accuracy. Importantly, **all models were provided with the same input information**, including solid types and load data. For domain-wise models, we treated **all deformable solids, rigid solids, and loads** as mesh points and concatenated them into a single input sequence. For graph-wise models,  we generated mesh edges using the **same kNN parameters** as Unisoma, ensuring consistency.
>
> We provide more results below (batch size: 1 for deforming plate and 50 for others). It is worth noting that **the advantages of Unisoma become more evident as the number of solids and points increases**, due to its modular explicit modeling.
>
> |||deforming plate|||cavity grasping|||tissue manipulation||
> |:-:|:-:|:-:|:-:|-|:-:|:-:|:--:|:-:|:-:|
> ||Param(M)|Time(s)|Mem(G)|Param(M)|Time(s)|Mem(G)|Param(M)|Time(s)|Mem(G)|
> |GINO|1.41|49.86|2.47|1.41|4.37|12.29|1.41|1.89|4.52|
> |GNO|1.14|29.81|7.64|1.23|8.44|23.45|1.23|1.73|5.53|
> |OFormer|1.48|48.55|0.92|1.48|4.16|7.41|1.48|1.14|1.80|
> |ONO|1.31|50.96|0.59|1.65|2.24|3.36|1.65|1.13|0.98|
> |Transolver|1.44|67.87|0.72|1.44|3.08|3.75|1.44|1.79|1.05|
> |Unisoma|0.92|50.20|0.41|1.40|1.92|0.86|1.40|1.45|0.63|
>
> > "implicit" and "explicit"
>
> We first confirm that all models received the **same inputs**, except for edges, which were used only where supported. The distinction between "explicit" and "emplicit" we defined lies **not in the input itself**, but in **how the model structure leverages and organizes that information**. For example, Transolver concatenates all objects as a single domain-level sequence input and learns interactions implicitly via attention. The model does not explicitly structure the pairwise physical relations (e.g., contact constraints or force) that drive deformation.
>
> In contrast, **Unisoma adopts an explicit modeling paradigm**, where physical interactions are structurally represented in the model architecture: the **contact modules** handle potential contacts, the **adaptive interaction allocation** computes equivalent load and constraint, and the **deformation triplet** encodes their influence on deformation. This structured design decomposes dynamics into controllable components. We will clarify this distinction more clearly in the revision.
>
> > Writing issues
>
> Thanks again for insightful suggestions and we will revise accordingly.
>
> Finally, as stated in the *Software and Data*  (Line 448), we will open data and code upon acceptance to ensure the reproducibility  and advance the field.

---

### Decision · Program_Chairs · 2025-05-01

**Decision:**

Accept (poster)

**Comment:**

This paper presents a transformer-based framework for explicitly modeling multi-solid interaction. Four reviewers recommended accepting this work. AC agrees that this work is interesting and deserves to be published on ICML 2O25. The reviewers did raise some valuable concerns that should be addressed in the final camera-ready version of the paper.